# Single cell, super-resolution imaging reveals an acid pH-dependent conformational switch in SsrB regulates SPI-2

Andrew Tze Fui Liew[1†], Yong Hwee Foo[1†], Yunfeng Gao[1], Parisa Zangoui[1], Moirangthem Kiran Singh[1], Ranjit Gulvady[1‡], Linda J Kenney[1,2*]

[1]Mechanobiology Institute, T-Lab, National University of Singapore, Singapore, Singapore; [2]Biochemistry and Molecular Biology, University of Texas Medical Branch, Galveston, United States

**Abstract** After *Salmonella* is phagocytosed, it resides in an acidic vacuole. Its cytoplasm acidifies to pH 5.6; acidification activates pathogenicity island 2 (SPI-2). SPI-2 encodes a type three secretion system whose effectors modify the vacuole, driving endosomal tubulation. Using super-resolution imaging in single bacterial cells, we show that low pH induces expression of the SPI-2 SsrA/B signaling system. Single particle tracking, atomic force microscopy, and single molecule unzipping assays identified pH-dependent stimulation of DNA binding by SsrB. A so-called phosphomimetic form (D56E) was unable to bind to DNA in live cells. Acid-dependent DNA binding was not intrinsic to regulators, as PhoP and OmpR binding was not pH-sensitive. The low level of SPI-2 injectisomes observed in single cells is not due to fluctuating SsrB levels. This work highlights the surprising role that acid pH plays in virulence and intracellular lifestyles of *Salmonella*; modifying acid survival pathways represents a target for inhibiting *Salmonella*.

DOI: https://doi.org/10.7554/eLife.45311.001

*For correspondence:
likenney@utmb.edu

[†]These authors contributed equally to this work

Present address: [‡]Institut, CurieParis, France

Competing interests: The authors declare that no competing interests exist.

## Introduction

*Salmonella enterica* serovar Typhimurium is a pathogen that causes gastroenteritis in humans and a typhoid-like disease in the mouse. *Salmonella* pathogenicity is largely conferred by the presence of horizontally-acquired virulence genes encoded within genomic regions called *Salmonella* pathogenicity islands (SPIs). The most well characterized genomic islands are SPI-1 and SPI-2, which encode two distinct type-three secretion systems (T3SS), as well as genes encoding secreted effectors that are important for pathogenesis (*Hensel, 2000*; *Lee et al., 1992*). The SPI-1 T3SS aids in the initial attachment and invasion of the intestinal epithelium (*Zhou and Galán, 2001*), while SPI-2 genes play an essential role in survival of *Salmonella* within the macrophage vacuole and its subsequent maturation into a *Salmonella*-containing vacuole (SCV) (*Feng et al., 2003*; *Garmendia et al., 2003*; *Lee et al., 2000*). Interestingly, expression of SPI-1 regulators is bistable, and is suggested to be the result of either a 'hedge-betting' or 'division of labour' strategy to ensure bacterial survival in the host (*Arnoldini et al., 2014*). Whether expression of SPI-2 genes shows similar phenotypic variation has not been fully investigated. Previous reports showed that in vitro, SPI-2 is only detected in a minority of the population (~13%) (*Chakraborty et al., 2015*). Typically, there are one or at most two injectisomes/cell, most often they are located at the cell pole (*Chakraborty et al., 2015*; *Chakravortty et al., 2005*).

Regulation of the SPI-2 pathogenicity island is complex and involves silencing by the nucleoid associated protein H-NS (*Lucchini et al., 2006*; *Gao et al., 2017*; *Liu et al., 2010*; *Winardhi et al.,*

**eLife digest** *Salmonellae* are a group of bacteria that can cause vomiting and diarrhea if we consume contaminated food. Once in the bowel, the bacteria get inside our cells, where they stay in a compartment called the vacuole. This environment is very acidic, and the inside of the microbes also becomes more acidic in response. This change helps *Salmonella* to switch on genes that allow them to survive and infect humans, but it is still unclear how this mechanism takes place.

To investigate this question, Liew, Foo et al. harnessed a recent technique called super-resolution imaging, which lets scientists see individual molecules in a cell. First, the technique was used to count a protein called SsrB as well as the enzyme that activates it, SsrA. The role of SsrB is to bind to DNA and turn on genes involved in making proteins that help *Salmonella* thrive. These studies revealed that the levels of SsrA/B proteins increased three-fold in an acidic environment.

Then, Liew, Foo et al. followed SsrB inside cells, knowing that fast-moving particles are free in solution, while slow-moving particles are typically bound to DNA. In acidic conditions, the proportion of SsrB bound to DNA doubled. Finally, further experiments revealed that when the environment was acidic, SsrB became five times more likely to bind to DNA. Taken together, the results suggest that acidic conditions trigger a cascade of events which switch on genetic information that allows *Salmonella* to survive.

If SsrB could be prevented from responding to acid stress, it could potentially stop *Salmonella* from surviving inside host cells. This knowledge should be applied to drive new treatment strategies for *Salmonella* and other microbes that infect human cells.

DOI: https://doi.org/10.7554/eLife.45311.002

*2015*; *Navarre et al., 2006*) and anti-silencing by response regulators (RRs) (*Desai et al., 2016*; *Walthers et al., 2011*; *Will et al., 2014*). RRs are part of a signal transduction system prevalent in bacteria. Such two-component systems consist of a membrane-bound histidine kinase (HK) and a cytoplasmic RR, which binds to DNA and activates gene transcription. The SsrA/B two-component system plays a crucial role in regulating SPI-2 gene expression (*Feng et al., 2003*; *Garmendia et al., 2003*; *Lee et al., 2000*; *Feng et al., 2004*) (see *Kenney, 2018* for a review and see *Figure 1*). Activation of SPI-2 genes requires phosphorylation of the RR SsrB on a conserved aspartic acid residue by its kinase SsrA (*Feng et al., 2004*). Upon activation, SsrB binds to AT-rich regions of DNA and activates transcription of SPI-2 promoters via displacement of the nucleoid-binding protein H-NS (*Walthers et al., 2011*) (*Figure 1*, left), as well as direct recruitment of RNA polymerase (*Walthers et al., 2007*). The expression of *ssrAB* is surprisingly complex; a promoter for *ssrB* resides in the coding region of *ssrA*, a 30 bp intergenic region lies between *ssrA* and *ssrB,* and both genes have extensive untranslated regions (*Walthers et al., 2007*), suggesting post-transcriptional or translational control (see *Figure 2A*). By comparison, in SPI-1, the unusually long untranslated region of the *hilD* mRNA functions as a hub for diverse mechanisms of post-transcriptional regulation (*Golubeva et al., 2012*). Each component of the enigmatic SsrA/B two-component system is regulated by separate global regulators EnvZ/OmpR (*Feng et al., 2003*; *Lee et al., 2000*) and PhoQ/P (*Bijlsma and Groisman, 2005*), indicating an uncoupling of the operon. In vitro transcription experiments demonstrate OmpR~P stimulation of *ssrA* and PhoP~P activation of *ssrB* (this work). This complexity was confounding, but recent studies demonstrated a non-canonical role for unphosphorylated SsrB in the absence of its kinase SsrA in driving biofilm formation and establishment of the carrier state (*Desai et al., 2016*), indicating a dual function for SsrB in controlling *Salmonella* lifestyles (*Figure 1*, right). In the present work, we count SsrA and SsrB molecules using photoactivation localization microscopy (PALM) and demonstrate their uncoupling and stimulation by acid pH. This complex hierarchy of gene activation ensures that activation of SPI-2 occurs only under conditions that presumably mimic the macrophage vacuole such as low pH, low Mg$^{2+}$ (*Hensel, 2000*; *Chakraborty et al., 2015*) and high osmolality (*Chakraborty et al., 2015*).

Upon encountering the acidic environment of the SCV, *Salmonella* acidifies its cytoplasm in an OmpR-dependent manner through repression of the *cadC/BA* system (*Chakraborty et al., 2015*). Intracellular acidification provides an important signal for expression and secretion of SPI-2 effectors. There is now increasing evidence that this change in intracellular pH is important for pathogenesis

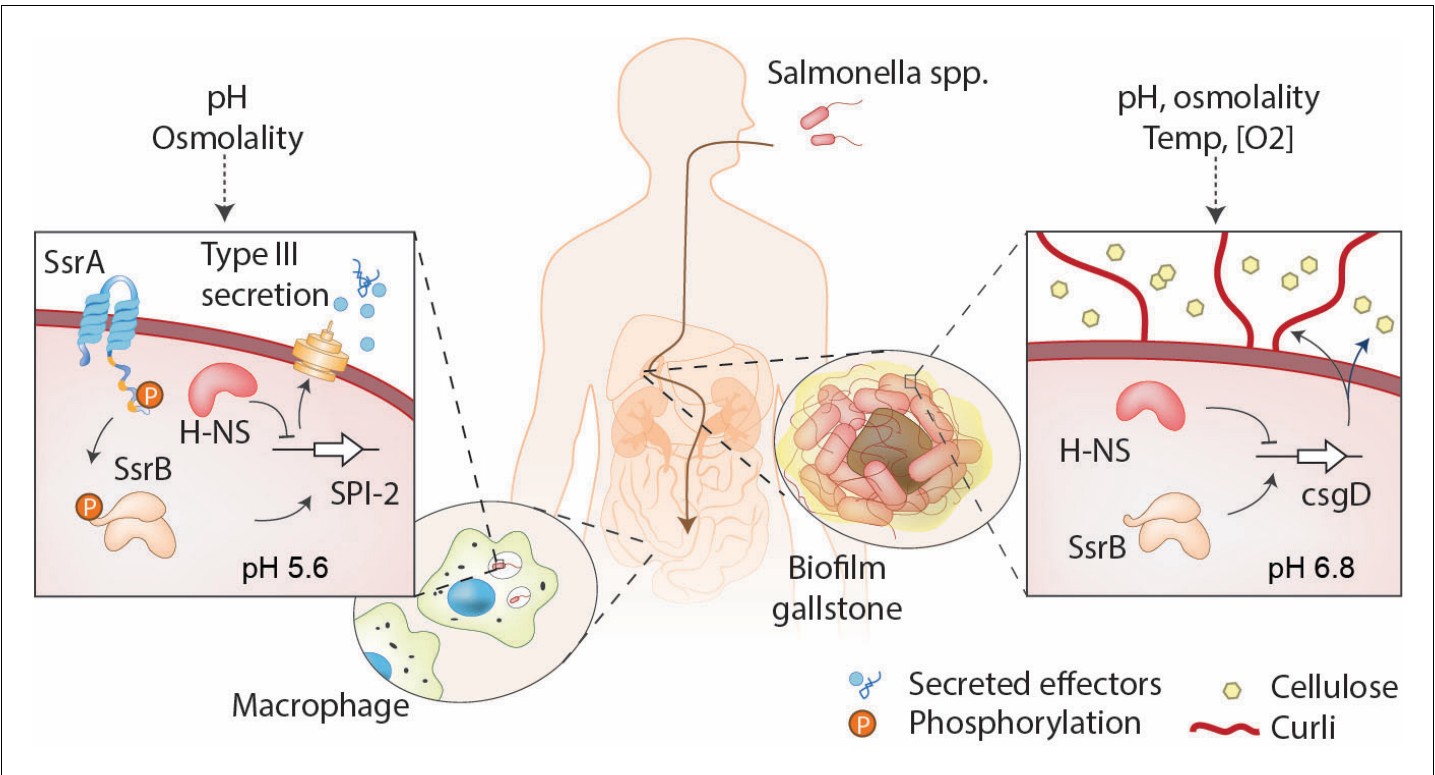

**Figure 1.** SsrB is a lifestyle switch that can function non-canonically. Left: When *Salmonella* resides in an acidic compartment such as the macrophage vacuole, its cytoplasm acidifies to pH 5.6 through the action of EnvZ/OmpR (*Chakraborty et al., 2015*; *Chakraborty et al., 2017*). In response to acidification, the number of molecules of SsrA and SsrB increases, as does SsrB binding to DNA (this work). SsrB~P functions to de-repress H-NS at SPI-2 (*Walthers et al., 2011*) and to activate SPI-2 transcription (*Feng et al., 2004*). Right: At neutral pH (pH$_i$ = 6.8), the SsrA kinase is nearly absent, and unphosphorylated SsrB de-represses H-NS at the *csgD* promoter, the master regulator of biofilms, driving biofilm expression (*Desai et al., 2016*). This promotes the carrier state, as *Salmonella* forms biofilms on gallstones in the gall bladder.

DOI: https://doi.org/10.7554/eLife.45311.003

(*Chakraborty et al., 2015*; *Chakraborty et al., 2017*; *Choi and Groisman, 2016*), although little is known as to how cytoplasmic acidification leads to SPI-2 gene activation. In particular, the effect of acidification on SsrA and SsrB has not been thoroughly investigated.

We therefore sought to characterize the response of SsrA and SsrB to acid pH in *Salmonella* using fixed and live cell imaging methods. We constructed functional, chromosomal, photoactivatable mCherry (PAmCherry) fusions to SsrA and SsrB and characterized their response to acid and neutral pH. Using PALM, in single cells we observed that the levels of SsrA and SsrB were significantly increased in acid pH compared to neutral pH. Single-particle tracking-PALM (Spt-PALM) demonstrated a concomitant increase in SsrB binding to DNA in live cells. Acid pH increased the cooperative binding of SsrB to DNA in vitro, as evident by atomic force microscopy (AFM), suggesting that acid pH induces a conformational change in SsrB, which increases its affinity for DNA. A 5-fold increase in DNA binding affinity was identified using a single molecule unzipping assay (*Gulvady et al., 2018*). Together, our results identify an important role for acid pH in regulating SPI-2 gene activation by controlling the levels of SsrA and SsrB, as well as regulating the affinity of SsrB for DNA.

## Results

### Construction of active SsrA and SsrB photoactivatable fusions

In order to characterize the effect of acid pH on SsrB and SsrA function, we determined the number of SsrB and SsrA molecules using PALM imaging of *Salmonella* grown under SPI-2-inducing (pH 5.6)

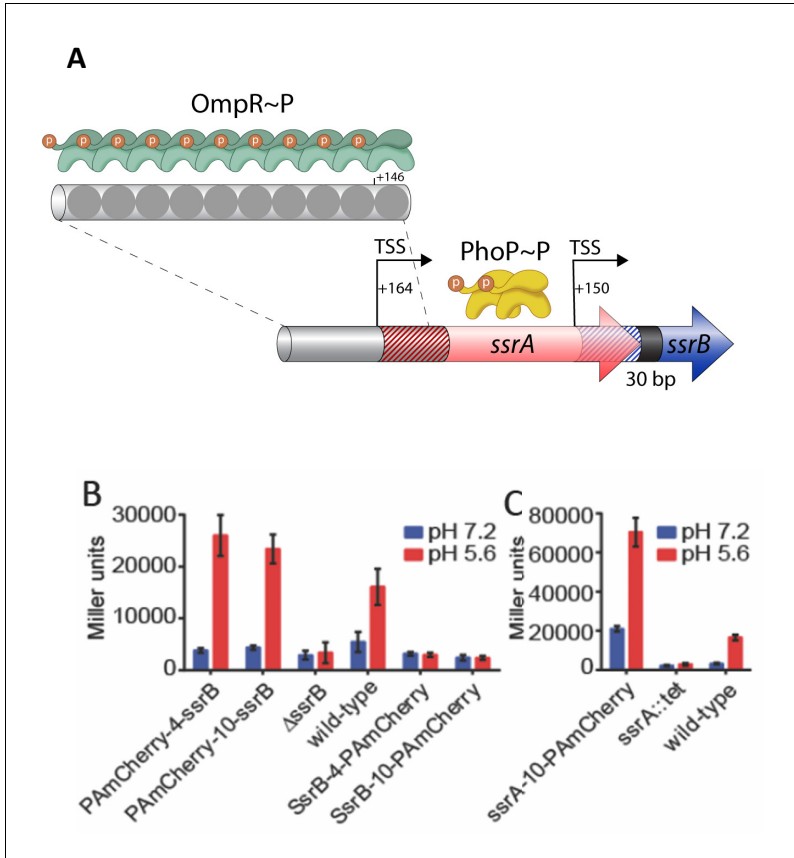

**Figure 2.** SsrB- and SsrA-PAmCherry fusions induce SPI-2 promoter activation in response to acid pH. (**A**) The *ssrA/B* gene structure. OmpR binds upstream of *ssrA*, PhoP binds upstream of *ssrB*. The transcription start sites are noted by the bent arrows and the untranslated regions are denoted by the cross-hatchings. See *Feng et al.* *(2003)* for more details. (**B**) The ability of N-terminal fusions of SsrB linked via a 4XGGSG or 10XGGSG linker to PAmCherry (PAmCherry-4-SsrB and PAmCherry-10-SsrB) or C-terminal fusions (SsrB-4-PAmCherry and SsrB-10-PAmCherry) to activate transcription of a SPI-2-linked promoter P*sseI*-lacZ, was measured by a β-galactosidase assay. Both N-terminal SsrB fusions activated *sseI* transcription, indicating that the N-terminal fusions were functionally active. In contrast, the activity of SsrB C-terminal fusions was similar to a Δ*ssrB* strain, indicating that they were not functionally active. (**C**) A C-terminal fusion of SsrA linked via a 10XGGSG linker to PAmCherry (SsrA-10-PAmCherry) had higher activation of *sseI-lacZ* compared to the wild-type, but showed a similar fold induction between acid and neutral pH (3.4-fold and 5-fold). Error bars represent standard deviations obtained from three, independent experiments, each measurement was in triplicate.

DOI: https://doi.org/10.7554/eLife.45311.004
The following source data is available for figure 2:

**Source data 1.**
DOI: https://doi.org/10.7554/eLife.45311.005
**Source data 2.**
DOI: https://doi.org/10.7554/eLife.45311.006

and SPI-2 non-inducing conditions (pH 7.2). We used a photoactivatable (PA) fluorescent protein, PAmCherry, which has minimal blinking (*Durisic et al., 2014*). During PALM imaging, proteins tagged with PAmCherry undergo photoactivation events. Each individual photoactivation event ideally represents a single molecule, which allows the counting of cellular proteins (*Endesfelder et al., 2013*; *Uphoff et al., 2013*). We constructed C-terminal and N-terminal fusions of SsrB and SsrA with PAmCherry (see Materials and methods for more details). To establish that the SsrB and SsrA fusions were functional, we examined the ability of the fusions to activate transcription of *sseI-lacZ*, an SsrB-dependent gene (*Feng et al., 2004*). We measured the β-galactosidase activity of *sseI-lacZ* under SPI-2-inducing and non-inducing conditions in various *Salmonella* strains containing SsrB- and SsrA-

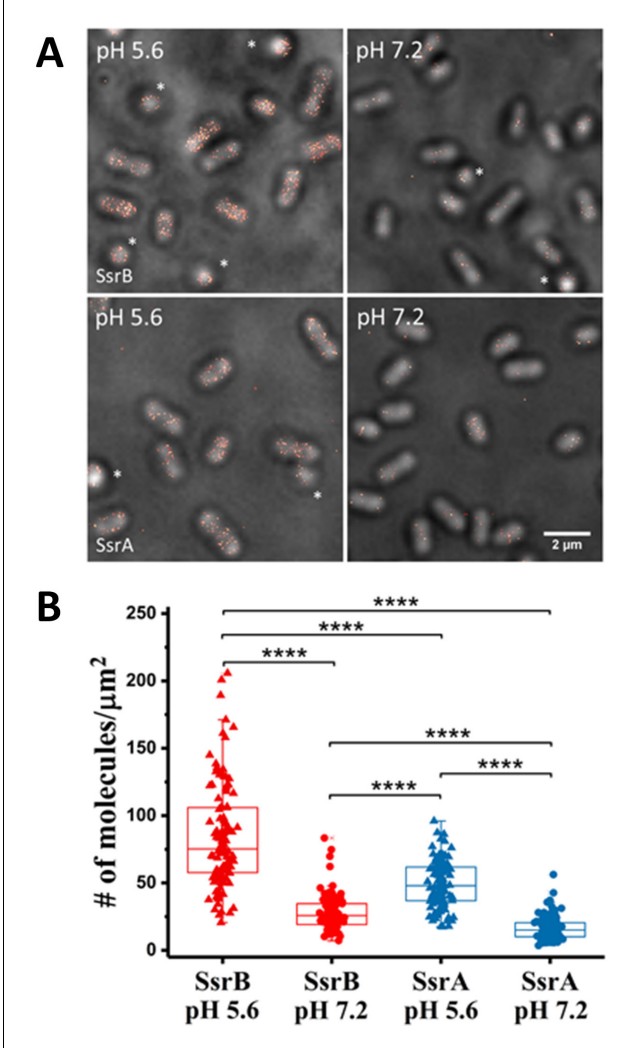

**Figure 3.** PALM imaging of SsrB and SsrA. (**A**) PALM-Brightfield overlay images of PAmCherry-SsrB (top panel) and SsrA-PAmCherry (bottom panel) grown in MgM at acid pH 5.6 or neutral pH 7.2. Asterisk highlights cells that are oriented axially to the glass coverslip. Scale bar = 2 μm. (**B**) Boxplot quantification of the number of SsrB (red) and SsrA (blue) molecules in individual cells. SsrB and SsrA PAmCherry levels were 3-fold higher during acid pH induction (triangles) compared to neutral pH (circles). The number of molecules/μm² was calculated by normalizing the total number of localizations within individual cells to its corresponding cell area (μm²). Results were combined from two, independent experiments. The total number of cells analyzed for SsrB at pH 5.6 = 120, SsrB at pH 7.2 = 92, SsrA at pH 5.6 = 117 and SsrA at pH 7.2 = 125. Statistical significance was determined by a two-tailed *t*-test (unpaired, unequal variances) using Microsoft Excel. **** Denotes $p<0.0001$.
DOI: https://doi.org/10.7554/eLife.45311.007

The following source data and figure supplement are available for figure 3:

**Source data 1.**
DOI: https://doi.org/10.7554/eLife.45311.009
**Figure supplement 1.** Two color, PALM-PAINT imaging of PAmCherry-SsrB (top panel) and SsrA–PAmCherry fusions (bottom panels) in red and cell membranes labeled with Nile red (green) grown in acid-inducing (left panels) or neutral pH conditions (right panels).
DOI: https://doi.org/10.7554/eLife.45311.008

PAmCherry fusions. *Salmonella* cells containing C-terminal SsrB fusions exhibited very low β-galacto-sidase activity during acid induction, similar to the Δ*ssrB* strain, indicating that these fusions were inactive (***Figure 2B***). In contrast, the N-terminal fusions displayed comparable, albeit slightly higher

activity (~1.5–1.6 fold), compared to wildtype. Importantly, these fusions showed similarly low β-galactosidase activity compared to wildtype cells when grown in neutral pH, consistent with previous reports that acid pH was required for transcription of SPI-2 genes (*Chakraborty et al., 2015*; *Feng et al., 2004*; *Beuzón et al., 1999*).

*sseI* promoter activity was also dependent on the kinase *ssrA*, as apparent by the low β-galactosidase activity of an *ssrA::tetRA* mutant (*Figure 2C*). The β-galactosidase activity of the C-terminal SsrA fusion was higher than the wildtype strain in both acid and neutral pH (4.2-fold and 6.5-fold, respectively, *Figure 2C*). Despite its higher activity, both the SsrA fusion and the wildtype protein exhibited a similar fold-induction when grown in acid compared to neutral pH (5.1- and 3.4-fold, respectively). These results indicate that the SsrA fusion responds to acid pH and activates *sseI* transcription in a similar manner as the wildtype strain. Taken together, our results from the *sseI-lacZ* assay indicate that N-terminal SsrB fusions and C-terminal SsrA fusions were functionally active and capable of activating SPI-2 gene transcription during acid induction.

## Acid pH increases SsrB and SsrA levels

Having shown that the SsrB and SsrA fusions could activate transcription of an SsrB-dependent promoter, we visualized the localization of both proteins when *Salmonella* cells were grown in acid or neutral pH using PALM imaging (*Figure 3A*, *Figure 3—figure supplement 1*). To quantify the number of molecules within each cell, we employed the LocAlization Microscopy Analyzer (LAMA) program to convert PALM localizations into copy numbers (*Malkusch and Heilemann, 2016*) (*Figure 3B*). SsrB and SsrA levels were, on average, about 3-fold higher during acid induction when compared to cells grown at neutral pH ($84 \pm 38$ molecules/$\mu m^2$ vs $28 \pm 13$ molecules/$\mu m^2$ for SsrB), and SsrA levels were always substantially lower than SsrB ($49 \pm 17$ molecules/$\mu m^2$ vs $16 \pm 8$ molecules/$\mu m^2$ for SsrA). The increase in both the SsrB RR and the SsrA HK suggested that increasing the concentration of both components of this two-component regulatory system was an important requirement for transcriptional activation of SPI-2 genes during acid stress (*Feng et al., 2003*).

We also observed differences in the levels of SsrB compared to SsrA (*Figure 3A*). For example, during acid induction, SsrB levels were slightly higher than SsrA (1.7-fold) and the variation in the levels of SsrB between individual cells was greater as compared to SsrA levels. This was reflected in the larger standard deviation of SsrB compared to SsrA at pH 5.6 (e.g. 38 vs 17). These differences were also consistent with previous reports that *ssrA* and *ssrB* transcription was uncoupled from one another (*Feng et al., 2004*).

We next examined the role of PhoP and OmpR in regulating SsrB levels, as binding sites for both proteins have been shown to be present upstream of the *ssrB* open reading frame (*Feng et al., 2003*; *Bijlsma and Groisman, 2005*). SsrB levels were reduced in the absence of *phoP* and *ompR* (*Figure 4*), consistent with a role for both regulatory proteins in controlling SsrB expression at acid pH. SsrB levels were reduced 5.3-fold (84 vs 16) in the *phoP::kan* mutant compared to wild-type, while in the *ompR::kan* mutant, SsrB was reduced only 1.8-fold (84 vs 47). These results suggest that PhoP plays a greater role than OmpR in regulating SsrB expression from the *ssrB* promoter during SPI-2 induction in vitro (see Discussion). In previous studies, SsrB-FLAG levels were not detected by Western blot in a *phoP* null background using a FLAG antibody (*Bijlsma and Groisman, 2005*). Our results demonstrate that super-resolution imaging is more sensitive than immunoblotting, because we could observe and count SsrB molecules in a *phoP* null background. This level of SsrB was 4-fold above the background limit determined in an *ssrB* null strain with no PAmCherry present (*Figure 4—figure supplement 1*). Expressing *phoP* or *ompR in trans* restored SsrB levels as well as acid induction (*Figure 4—figure supplement 2*).

## The low number of SPI-2 injectisomes on the cell surface is not due to single cell variation in SsrB levels

Unlike many other T3SSs, the *Salmonella* SPI-2 T3SS is not abundant, and is often localized to the cell pole (*Chakraborty et al., 2015*; *Chakravortty et al., 2005*). We visualized these SPI-2 appendages with immunofluorescence microscopy using rabbit antibodies raised against the translocon protein SseB and anti-rabbit antibodies conjugated with an Alexa-488 fluorophore in wild-type cells. The SPI-2 injectisome was only present on 17% of cells under optimal in vitro SPI-2 inducing conditions (*Figure 5A* upper panels), consistent with previous observations (*Chakraborty et al., 2015*).

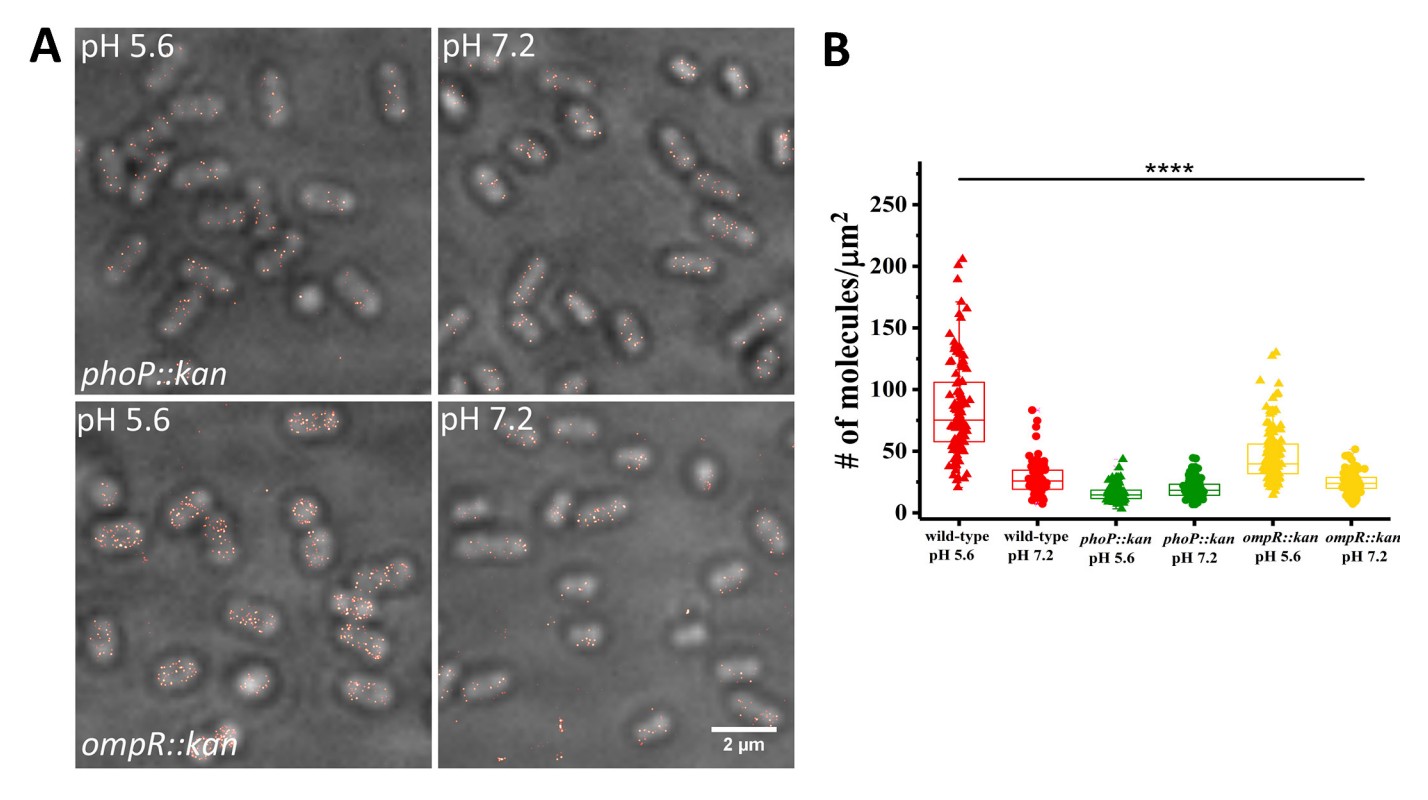

**Figure 4.** SsrB levels are reduced in *ompR* and *phoP* null strains. (**A**) PALM-Brightfield overlay images of cells expressing PAmCherry-SsrB in *phoP::kan* (top) or *ompR::kan* (bottom) strains. Cells were grown in MgM media in acid (pH 5.6) and neutral pH (pH 7.2). Scale bar is 2 µm. (**B**) Boxplot showing quantification of the number of PAmCherry-SsrB molecules in the *phoP::kan* and *ompR::kan* mutants when grown in acid pH or neutral pH. The # molecules/µm$^2$ was calculated by normalizing the total number of localizations within individual cells to its corresponding cell area (µm$^2$). Results were combined from two, independent experiments. The total number of cells analyzed for wild-type at pH 5.6 = 120, at pH 7.2 = 92, *phoP::kan* at pH 5.6 = 131, at pH 7.2 = 140, *ompR::kan* at pH 5.6 = 131, at pH 7.2 = 117 cells. To determine the background level, we imaged an *ssrB* null strain lacking PAmCherry. The average localization counts for the strain lacking PAmCherry was four localizations/µm$^2$ (n = 66 cells), while there were 16 localizations/µm$^2$ in the PAmCherry-SsrB *phoP::kan* strain. Thus, SsrB is detectible above the background. Statistical significance was determined by a two-tailed *t*-test (unpaired, unequal variances) using Microsoft Excel. The line above the bar graph indicates statistical significance between pairwise group comparisons of all groups. **** Denotes p<0.0001.
DOI: https://doi.org/10.7554/eLife.45311.010

The following source data and figure supplements are available for figure 4:

**Source data 1.**
DOI: https://doi.org/10.7554/eLife.45311.019
**Source data 2.**
DOI: https://doi.org/10.7554/eLife.45311.020
**Source data 3.**
DOI: https://doi.org/10.7554/eLife.45311.021
**Figure supplement 1.** Determining the background counts in an *ssrB* null strain in the absence of PAmCherry.
DOI: https://doi.org/10.7554/eLife.45311.011
**Figure supplement 2.** The *ompR::kan* and *phoP::kan* mutant strains were complemented by the expression of OmpR and PhoP from *p-ompR* and *p-phoP* plasmids *in trans*.
DOI: https://doi.org/10.7554/eLife.45311.012
**Figure supplement 2—source data 1.**
DOI: https://doi.org/10.7554/eLife.45311.013
**Figure supplement 3.** Transcription of the *ssrA* and *ssrB* promoters is directly coupled to the PhoQ/PhoP and EnvZ/OmpR signaling pathways, respectively.
DOI: https://doi.org/10.7554/eLife.45311.014
**Figure supplement 3—source data 1.**
DOI: https://doi.org/10.7554/eLife.45311.015

*Figure 4 continued*

**Figure supplement 4.**
DOI: https://doi.org/10.7554/eLife.45311.016
**Figure supplement 4—source data 1.**
DOI: https://doi.org/10.7554/eLife.45311.017
**Figure supplement 5.** DNA binding by PhoP or OmpR is not acid-sensitive.
DOI: https://doi.org/10.7554/eLife.45311.018

Translocons were even less prevalent (~5%) in a Δ*ssrB* mutant (**Figure 5A**, lower panels), indicating that SsrB is an important regulator of SPI-2 translocon production (**Beuzón et al., 1999**). The reason for injectisome heterogeneity is unknown, but we wondered whether it might result from single cell variation in SsrB levels, since we had observed heterogeneity in the levels of SsrB in individual cells under SPI-2 inducing conditions using PALM imaging (**Figures 3** and **4**). Previous observations of bistability of SPI-1 transcriptional regulators (**Saini et al., 2010**) led us to question whether there was a correlation between the presence of SseB translocons and higher levels of SsrB in individual cells. To address this, we grew cells containing the PAmCherry-SsrB fusion in MgM at acid pH and labeled them with anti-SseB antibody (**Figure 5B**) to distinguish between cells that were SseB plus (injectisome positive) from cells that lacked SseB (injectisome negative). We then quantified SsrB molecules in cells from both populations to determine if the SseB plus cells contained a higher number of SsrB molecules (**Figure 5BC**). On average, the number of SsrB molecules in the SseB plus cells was slightly higher than the SseB minus cells (134 vs 112 molecules/μm$^2$), but there was significant overlap in the distribution of SsrB levels in both populations. In other words, most of the SseB plus cells contained similar levels of SsrB as the SseB minus cells. Our results thus suggest that in vitro, SsrB levels do not exclusively determine translocon (injectisome) numbers and other processes must contribute to the variability.

## SsrB binding to DNA is acid-dependent

Because SPI-2 genes were acid-induced (**Chakraborty et al., 2015**; **Feng et al., 2004**; **Chakraborty et al., 2017**), we reasoned that acid pH might be an important factor in regulating the DNA binding dynamics of SsrB. To examine this, we used Spt-PALM to quantify DNA binding of SsrB under different growth conditions in live bacterial cells (**Figure 6**). We first grew *Salmonella* containing the PAmCherry-SsrB fusion in acid and neutral pH, and then placed these cells on agarose pads reconstituted with media at the appropriate pH. The power of the 405 nm activation laser was controlled such that only a single molecule was activated per cell. The 561 nm excitation laser tracked the single molecule. The location of single PAmCherry-SsrB molecules were followed until they photobleached and then they were linked to form tracks (**Figure 6A**). The apparent diffusion coefficients, $D$, were obtained from the distribution of displacement $r$ (the distance moved by the molecule in subsequent camera frames) plotted as a cumulative distribution function (CDF) (see **Figure 6—figure supplement 1**) (**Yang et al., 2016**; **Schütz et al., 1997**). A three-component diffusion model best fit our data (**Figure 6—figure supplement 1** and **Table 1**) (**Gao et al., 2017**). From the CDF fits, apparent $D$ values fell into three categories, (i) < 0.1 μm$^2$s$^{-1}$, (ii) 0.1–0.4 μm$^2$s$^{-1}$ and (iii) > 1.2 μm$^2$s$^{-1}$, representing (i) bound, (ii) transiently bound and (iii) free SsrB molecules (**Gao et al., 2017**; **Elf et al., 2007**; **Sanamrad et al., 2014**; **Stracy et al., 2015**). $D$ values of fixed cells expressing PAmCherry-SsrB grown in acid were less than <0.1 μm$^2$s$^{-1}$, further validating a low $D$ of immobile (bound) molecules. A significantly higher population of SsrB molecules was bound in acidic conditions compared to neutral pH ($F_1$ = 28.1% vs 11.3%). It was also interesting to note that the diffusion coefficient for the 'transiently bound' SsrB population ($D_2$) was significantly slower in acid compared to neutral pH, with values of $D_2$ = 0.14 μm$^2$s$^{-1}$ ($F_2$ = 34.5%) vs 0.25 μm$^2$s$^{-1}$ (22.2%), indicating that acidic conditions increased the affinity of SsrB for DNA (see **Figures 7** and **8**).

Using Spt-PALM, we also characterized the effect of different SsrB substitutions on DNA binding. K179A, a DNA-binding mutant of SsrB (**Carroll et al., 2009**) demonstrated significantly reduced binding to DNA compared to the wild-type ($F_1$ = 11.3% vs 28.1%) in acid pH (see **Figure 6**). This level of binding by K179A was identical to the level of wildtype SsrB bound under non-inducing conditions and defined a background limit. DNA binding was slightly above background with the phosphorylation site mutant D56A ($F_1$ = 20.5% in acid vs 13.0% in neutral pH). Neither the D56A nor the

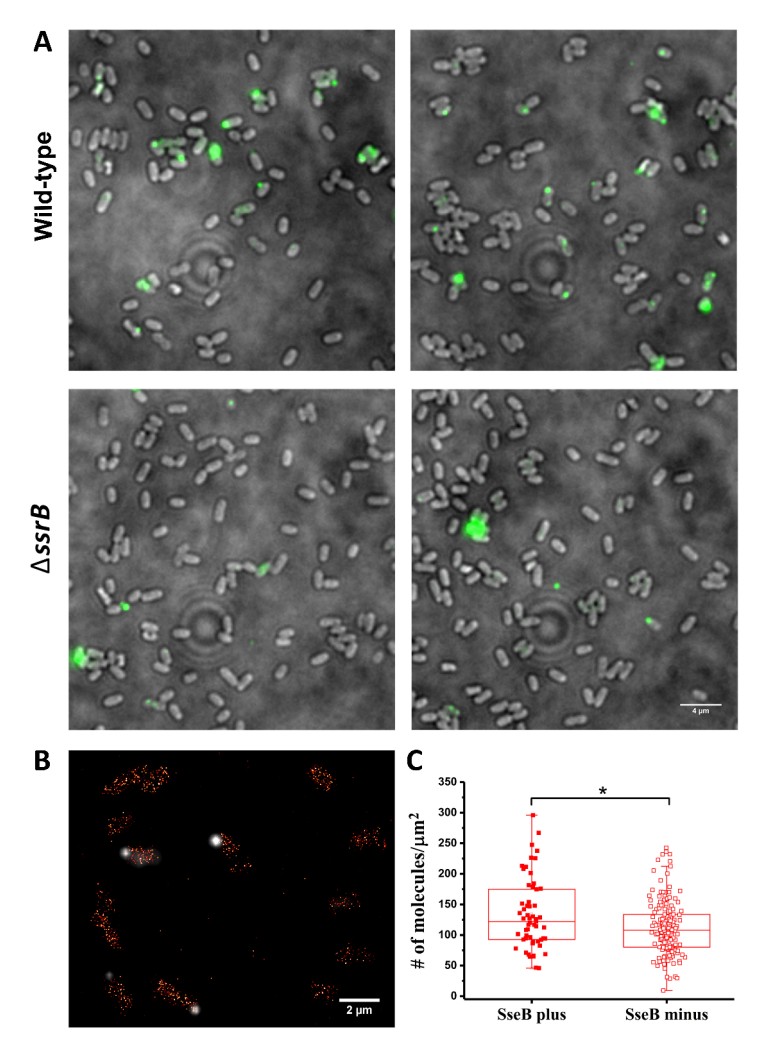

**Figure 5.** SsrB variation in vitro does not affect the appearance of SseB translocons on the cell surface. (**A**) SseB production requires SsrB. Immunofluorescence using rabbit anti-SseB antibody (green) of wild-type (top panels) and Δ*ssrB* cells (lower panels). In the wildtype, 17% of the population possessed injectisomes (of 276 cells examined), but only 5% of Δ*ssrB* cells (277 cells) were stained with SseB antibody. Two different images for each strain are shown. A and B are results combined from two, independent experiments. (**B**) PALM-SseB immunofluorescence overlay image of PAmCherry-SsrB. Arrows indicate cells that produce SseB on their surface. Scale bar is 2 μm. (**C**) Boxplot shows quantification of the number of PAmCherry-SsrB molecules in cells containing SseB on their surface (SseB plus) or without (SseB minus) (total = 227 cells). *Salmonella* was grown in MgM media at acid pH. The # molecules/μm$^2$ was calculated by normalizing the total number of localizations within individual cells to its corresponding cell area (μm$^2$). Results from two, independent experiments were combined. Statistical analysis was performed using an unpaired two-tailed *t*-test (Microsoft Excel). * Denotes $p < 0.05$.

DOI: https://doi.org/10.7554/eLife.45311.022

The following source data is available for figure 5:

**Source data 1.**
DOI: https://doi.org/10.7554/eLife.45311.023
**Source data 2.**
DOI: https://doi.org/10.7554/eLife.45311.024

K179A SsrB mutant was able to activate transcription of *sseI-lacZ*, a SPI-2 co-induced transcriptional fusion (*Figure 6E*), indicating that while the D56A mutant was not able to activate SPI-2 transcription, its elevated levels of DNA binding compared to K179A was most likely due to its binding to

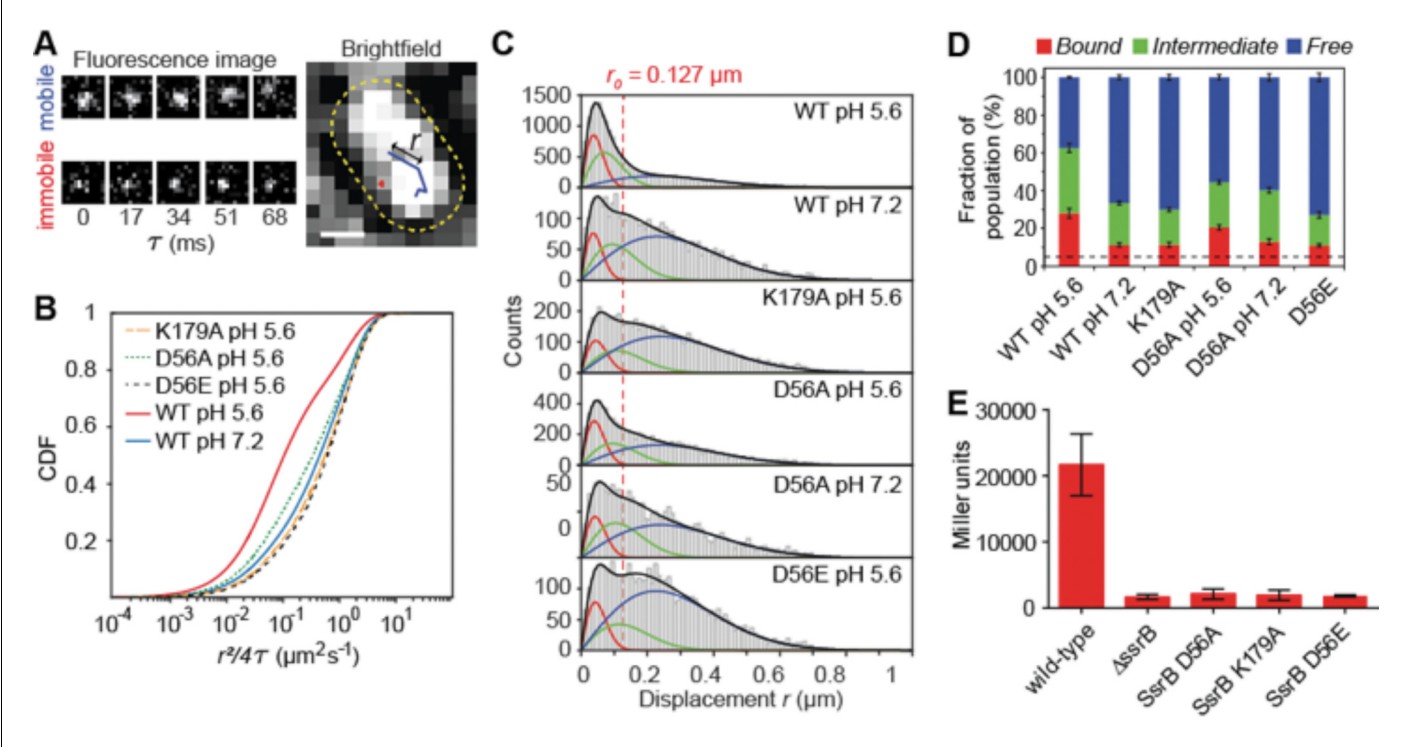

**Figure 6.** SsrB binding to DNA increases in acid pH. (**A**) The left panel contains two representative fluorescence signals of a mobile and immobile molecule. Images were acquired at 17 ms intervals. The right panel contains the tracks of the signals from the left panel. The yellow dotted line is the outline of the bacterial cell obtained from brightfield imaging segmentation. Displacement $r$ is the distance travelled in adjacent frames by a molecule. (**B**) The values of the displacement $r$, were plotted as a CDF. The CDF fits of various SsrB mutants and wild-type SsrB grown in different pH are shown. A shift in the curve to the right indicates an increase in $D$. The fitted values can be found in **Table 1**. (**C**) Values obtained from the CDF are represented onto the PDF histogram of $r$ to display the distribution of $r$ values responsible for the corresponding $D$. The red curve indicates the distribution for $D_1$, the green curve indicates the distribution for $D_2$ and the blue curve indicates the distribution for $D_3$. $r$ values below the red dotted line are threshold values indicative of the bound fraction obtained from fixed cells expressing SsrB (see **Table 1** for the number of tracks for each experiment and the number of cells analyzed). The distribution of $D_2$ extends past this threshold, suggesting that $D_2$ is a transient, weaker binding fraction. (**D**) Values of $F_1$, $F_2$ and $F_3$ of the SsrB mutants. An increase in the values of $F_1$ and $F_2$ for WT grown in pH 5.6 illustrates increased SsrB binding to DNA. The dotted line indicates the basal bound $F_1$ value (5.4%) obtained from PAmCherry alone, which is not bound to DNA. (**E**) SsrBD56A-, SsrBK179A-, and SsrBD56E-PAmCherry fusions do not activate SPI-2 gene transcription. Wild-type, Δ*ssrB* and strains expressing the corresponding PAmCherry-SsrB fusions containing *sseI-lacZ* were grown in acid MgM media and β-galactosidase activity was measured. The mean and standard deviations were obtained from two, independent experiments.

DOI: https://doi.org/10.7554/eLife.45311.025

The following source data and figure supplements are available for figure 6:

**Source data 1.**
DOI: https://doi.org/10.7554/eLife.45311.029
**Figure supplement 1.**
DOI: https://doi.org/10.7554/eLife.45311.026
**Figure supplement 2.** Structured illumination microscopy (SIM) images of *Salmonella* grown in pH 7.2 and 5.6.
DOI: https://doi.org/10.7554/eLife.45311.027
**Figure supplement 2—source data 1.**
DOI: https://doi.org/10.7554/eLife.45311.028

sites that do not require SsrB phosphorylation (*Desai et al., 2016*). This result illustrates that DNA binding alone by SsrB was insufficient for activating transcription.

**Table 1.** Apparent diffusion coefficient $D$ obtained from SptPALM of SsrB mutants.

| | No. of cells | No. of tracks | $F_1$ (%) | $F_2$ (%) | $F_3$ (%) | $D_1$ ($\mu m^2 s^{-1}$) | $D_2$ ($\mu m^2 s^{-1}$) | $D_3$ ($\mu m^2 s^{-1}$) |
|---|---|---|---|---|---|---|---|---|
| Fixed | 41 | 496 | 26.9 ± 4.0 | 65.0 ± 3.7 | 8.1 ± 0.6 | 0.020 ± 0.002 | 0.066 ± 0.003 | 0.88 ± 0.08 |
| WT pH5.6 | 89 | 2618 | 28.1 ± 2.6 | 34.5 ± 2.3 | 37.3 ± 0.5 | 0.041 ± 0.002 | 0.14 ± 0.01 | 1.49 ± 0.02 |
| WT pH7.2 | 71 | 569 | 11.3 ± 1.1 | 22.2 ± 1.0 | 66.5 ± 1.2 | 0.037 ± 0.004 | 0.25 ± 0.03 | 1.53 ± 0.03 |
| K179A pH5.6 | 113 | 1038 | 11.3 ± 1.3 | 18.7 ± 1.2 | 69.9 ± 1.5 | 0.056 ± 0.006 | 0.32 ± 0.05 | 1.70 ± 0.03 |
| D56A pH5.6 | 109 | 1345 | 20.5 ± 1.4 | 24.0 ± 1.1 | 55.5 ± 1.2 | 0.045 ± 0.003 | 0.26 ± 0.03 | 1.61 ± 0.03 |
| D56A pH7.2 | 74 | 384 | 13.0 ± 1.5 | 27.3 ± 1.4 | 59.8 ± 1.9 | 0.050 ± 0.005 | 0.31 ± 0.04 | 1.66 ± 0.04 |
| D56E pH5.6 | 111 | 796 | 11.1 ± 0.8 | 16.2 ± 1.7 | 72.8 ± 2.2 | 0.053 ± 0.004 | 0.39 ± 0.06 | 1.52 ± 0.03 |
| PAmCherry pH5.6 | 53 | 715 | 5.4 ± 0.3 | 27.8 ± 1.3 | 66.7 ± 1.4 | 0.029 ± 0.004 | 0.67 ± 0.04 | 3.61 ± 0.07 |
| PhoP pH 5.6 | 81 | 3665 | 9.8 ± 1.5 | 23.7 ± 1.3 | 66.4 ± 0.4 | 0.033 ± 0.004 | 0.14 ± 0.01 | 1.54 ± 0.01 |
| PhoP pH 7.2 | 103 | 2930 | 10.6 ± 0.6 | 18.8 ± 1.5 | 70.6 ± 1.8 | 0.043 ± 0.003 | 0.40 ± 0.04 | 1.58 ± 0.03 |
| OmpR pH 5.6 | 43 | 1610 | 21.9 ± 1.4 | 20.9 ± 1.0 | 57.0 ± 1.3 | 0.054 ± 0.003 | 0.26 ± 0.03 | 1.34 ± 0.02 |
| OmpR pH 7.2 | 55 | 2119 | 16.4 ± 1.0 | 28.0 ± 0.9 | 55.6 ± 1.0 | 0.045 ± 0.003 | 0.25 ± 0.02 | 1.45 ± 0.02 |
| OmpR pH 5.6 (*E.coli*) | 23 | 1644 | 24.3 ± 0.9 | 39.4 ± 3.7 | 36.2 ± 4.2 | 0.066 ± 0.003 | 0.55 ± 0.06 | 2.06 ± 0.14 |
| OmpR pH 7.2 (*E.coli*) | 22 | 478 | 15.3 ± 1.2 | 45 ± 1.8 | 39.7 ± 2.4 | 0.057 ± 0.005 | 0.40 ± 0.03 | 2.06 ± 0.09 |

DOI: https://doi.org/10.7554/eLife.45311.038

## A 'constitutively active' SsrB phosphomimetic (D56E) does not bind DNA in live cells

Interestingly, we also examined the DNA binding ability of a 'constitutively active' SsrB variant (D56E), which was previously reported to repress *hilA* expression in a mouse infection model (*Pérez-Morales et al., 2017*). In acidic pH, the binding of the D56E mutant was $F_1$ = 11.1% compared to wild-type SsrB (28.1%). This level of binding was similar to the background level of binding observed with the wildtype at neutral pH and the DNA binding mutant K179A (*Figure 6B*). In addition, the D56E mutant was unable to activate transcription of *ssel-lacZ* when grown in acid pH (*Figure 6E*). This result indicates that the presumed phospho-mimicry was not effective in supporting high affinity DNA binding by SsrB, and the effects on *hilA* in the mouse studies were likely the result of over-expression of SsrBD56E (*Pérez-Morales et al., 2017*) (see Discussion). Overall, our Spt-PALM results reveal that acid pH increases the binding of SsrB to DNA in live cells, and binding does not require phosphorylation. Furthermore, a substitution that was reported to mimic the phosphorylated state does not increase SsrB binding to DNA above background.

## Acid pH increases SsrB binding to DNA

Based on the Spt-PALM results, we considered the possibility that an increase in SsrB binding might be due to an intrinsic change in protein conformation, since *Salmonella* encounters an acid environment (pH 5.6) in the vacuole (*Chakraborty et al., 2015*). To test this possibility, we purified full length wild-type SsrB and examined binding to a 704 bp DNA fragment containing the promoter and regulatory region of the SPI-2 effector *sifA* using AFM. The naked DNA in the absence of protein is shown in *Figure 7A* (left panels). Consistent with previous reports (*Desai et al., 2016*), we observed that, even at low protein concentrations (30 nM), at pH 6.8, unphosphorylated SsrB could bind and bend at specific sites of the DNA (*Figure 7A*, lower panels), although at this concentration, only a low level of binding was evident (see below). At acid pH (pH 6.1, the intracellular pH we measured previously [*Chakraborty et al., 2017*]), a significant increase in SsrB binding to the *sifA* promoter was evident both in the AFM images (*Figure 7A*, upper panels) and was reflected in the extensive relative height distribution histogram compared to pH 6.8 (*Figure 7B*). This widespread condensation and cooperative binding of SsrB to DNA at acid pH was consistent with the increased binding observed in our Spt-PALM experiments (*Figure 6*), and also indicated that SsrB undergoes a

conformational change in acid pH, which increases its affinity for DNA. As expected, the DNA binding mutant K179A (*Carroll et al., 2009*) was unable to bind to DNA at either pH (*Figure 7A*).

We were able to measure the change in binding affinity produced by acid pH using a highly sensitive single molecule counting assay (*Gulvady et al., 2018*). In this assay, a DNA hairpin was created that contained an SsrB binding site (*Figure 8A*). Delayed hairpin unzipping was observed in the presence of SsrB (*Figure 8—figure supplement 1*). At neutral pH, the $K_D$ for SsrB binding was 239 nM (*Figure 8B*), whereas at pH 6.1, the $K_D$ was reduced by 5-fold to 47 nM (*Figure 8B*). Thus, acid pH not only increases the number of SsrB molecules in the cell, but it also leads to higher affinity binding to DNA. Elution of SsrB by size exclusion chromatography was identical at acid and neutral pH, indicating that SsrB dimerization was not pH-dependent (*Figure 8—figure supplement 2*).

## Is acid-dependent DNA binding an intrinsic property of RRs?

We were interested in determining whether acid-dependent DNA binding might be an hitherto unrecognized intrinsic property of RRs. An increase in DNA binding affinity in acid pH was observed with the RR OmpR, but the amount of OmpR bound to DNA was not determined (*Chakraborty et al., 2017*). We therefore used PALM and Spt-PALM to compare the localization and dynamics of OmpR and PhoP, two RRs that regulate SPI-2 (*Figure 4*), after growth in acid or neutral pH. Previous experiments have shown that OmpR (*Feng et al., 2003*) and PhoP (*Bijlsma and Groisman, 2005*) directly bind to the promoter region of *ssrB*, stimulating its transcription, and the phosphoproteins bind and activate transcription in vitro (*Figure 4—figure supplement 3*).

We constructed two PhoP-PAmCherry fusions, containing different linker lengths attached to PAmCherry (see Materials and methods). Both fusions were examined for their ability to activate transcription of an *ssrB-lacZ* transcriptional fusion. The activity of the PhoP-4-PAmCherry construct was ~60% compared to wild-type, while the PhoP-10-PAmCherry fusion was 69% active (*Figure 4—figure supplement 4A*). Similarly, an OmpR-PAmCherry fusion was constructed and assayed for function of *ompF* and *ompC* attached to GFP reporters (*Foo et al., 2015*). The OmpR-PAmCherry construct was 113% compared to wild-type at *ompF* (*Figure 4—figure supplement 4B*), and 54% compared to wild-type at *ompC* (*Figure 4—figure supplement 4C*), similar to what we observed in *E. coli* (*Foo et al., 2015*).

Using PALM, we visualized the PhoP-PAmCherry fusion and observed that overall levels of PhoP were about 2.3 and 3.5-fold higher than SsrB at acid and neutral pH, respectively. PhoP levels were also slightly induced in acid pH about 1.9-fold (189 vs 100 molecules/μm$^2$) (*Figure 4—figure supplement 5A–B*). In contrast, SsrB levels increased 3-fold in acid pH compared to neutral pH (*Figure 4*). Unlike SsrB, PhoP-PAmCherry binding to DNA did not increase in acid compared to neutral pH (*Figure 4—figure supplement 4C*), and we actually observed a slight increase in the apparent diffusion coefficient of the $F_2$ transiently bound, (0.14 μm$^2$s$^{-1}$ (23.7%) vs 0.4 μm$^2$s$^{-1}$ (18.8%); *Table 1*).

In contrast, OmpR levels in acidic media increased only 1.2-fold compared to neutral pH (183 ± 50 vs 153 ± 43 molecules/μm$^2$) (*Figure 4—figure supplement 4D–E*), but there was a slight increase in the DNA-bound fraction $F_1$ (21.9 ± 1.4) compared to neutral pH (16.4 ± 1.0) (*Figure 4—figure supplement 4F*). Thus, SsrB binding to DNA was exquisitely acid-sensitive, increasing ~60% (*Figures 5* and *6* and *Table 1*), whereas OmpR binding only increased 5% and PhoP binding did not change. These results suggest that increased DNA binding in acid pH is not an intrinsic property of RRs, but rather is due to an acid-dependent conformational change in SsrB.

## Acid pH does not relax the nucleoid

What is the stimulus during acid stress that increases SPI-2 gene expression? One model proposed that acid pH led to chromosome relaxation, exposing sites for OmpR that were not normally available (*Quinn et al., 2014*). That model was based on studies of plasmid DNA in the presence of novobiocin. To measure chromosome compaction directly, we imaged the area of the *Salmonella* nucleoid (*Gao et al., 2017*) stained with DRAQ5 after growth at different pH values. The area of the nucleoid in acid was actually smaller (0.50 ± 0.22 μm$^2$) than when grown at neutral pH (0.62 ± 0.15 μm$^2$). The ratio of the two was 0.81 (*Figure 6—figure supplement 2*). Thus, the nucleoid was more compact at acid pH, in keeping with previous observations (*Gao et al., 2017*; *Foo et al., 2015*). To eliminate the possibility that this difference was affected by the DNA replication rate, we compared the fluorescence intensity of the DRAQ5 stained nucleoid, which is proportional to the amount of

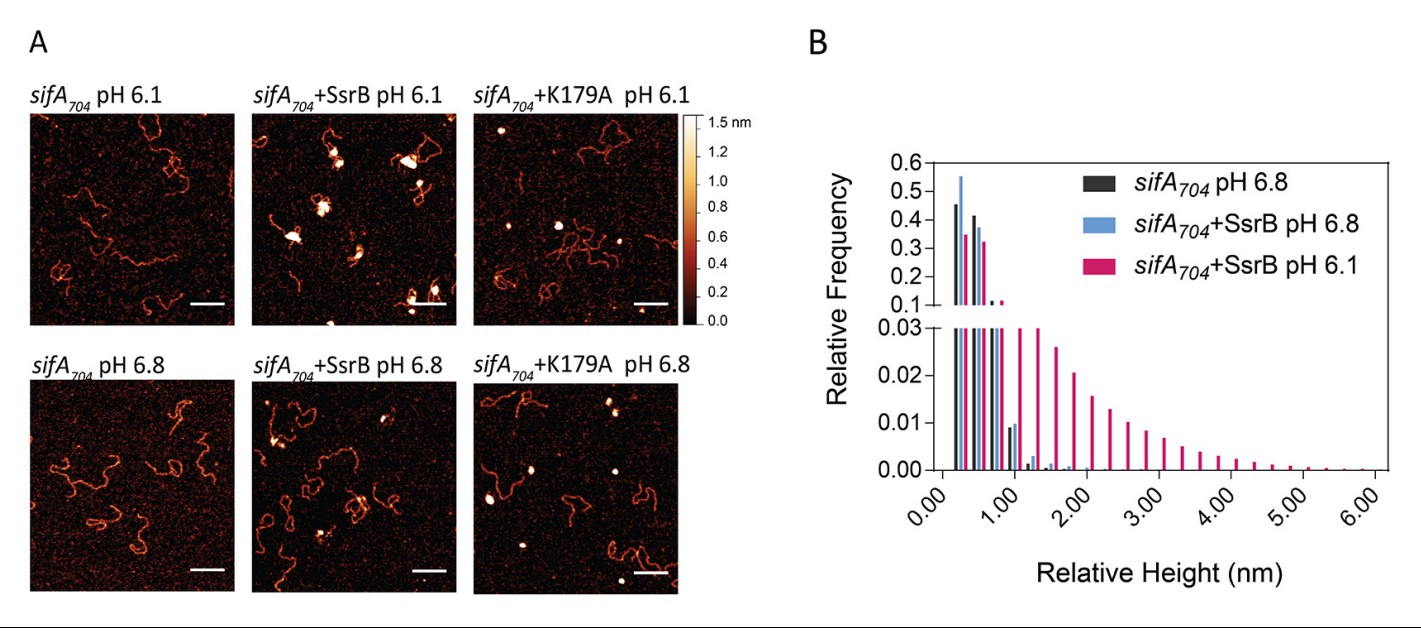

**Figure 7.** SsrB binding to DNA is increased in acid pH. (**A**) AFM image of *sifA₇₀₄* DNA in the absence of SsrB, or in the presence of 30 nM SsrB or 30 nM SsrBK179A at pH 6.8 (lower panels) and pH 6.1 (upper panels). The AFM images of *sifA₇₀₄* DNA were identical at pH 6.8 and pH 6.1 (left panels). When 30 nM SsrB was added, binding and bending of DNA was apparent in a minor percentage of the population at pH 6.8, whereas binding and DNA condensation was significantly increased at pH 6.1. The pH values were selected based on the previously measured pH values in response to acid stress (*Chakraborty et al., 2017*). The K179A mutant did not bind to DNA (*Carroll et al., 2009*). Scale bar, 100 nm. (**B**) A relative height distribution histogram of the *sifA₇₀₄* promoter complexed with 30 nM SsrB at pH 6.8 (blue bars) compared to pH 6.1 (red bars). The apparent height of naked DNA peaks at ~0.5 nm (black columns), while above 1 nm, the probability of the height decreases to near zero (*Gao et al., 2017*). Therefore, the height detected at above 1 nm corresponds to SsrB binding. SsrB shows an extended height distribution at pH 6.1 (up to 5 nm), reflecting its enhanced ability to bind, bend, and condense DNA at acid pH (*Desai et al., 2016*).

DOI: https://doi.org/10.7554/eLife.45311.030

The following source data is available for figure 7:

**Source data 1.**

DOI: https://doi.org/10.7554/eLife.45311.031

DNA/cell. The average intensity/cell in acid pH was 66410, while in neutral pH it was 80553. The ratio was 0.82, that is similar to the ratios of nucleoid area. Thus, DNA relaxation in acid pH was not apparent. The method was capable of discerning relaxation, because addition of the gyrase inhibitor novobiocin increased nucleoid area by 20% (*Gao et al., 2017*).

## Discussion

The system used to construct the photoactivatable fusion proteins employed in this study will be useful for future studies of transcriptional regulators in *Salmonella* and *E. coli*. In particular, PALM localization studies combined with Spt-PALM to track protein dynamics under different environmental conditions will enhance our understanding of the signaling repertoire in bacteria.

### Acid pH increased SsrA/B levels and SsrB affinity for DNA

Upon encountering the acidic environment of the SCV, *S.* Typhimurium undergoes intracellular acidification via repression of the *cadC/BA* system by OmpR (*Chakraborty et al., 2015*). This acidification step is essential for secretion of SPI-2 effectors such as SseJ (*Chakraborty et al., 2015*). However, how cytoplasmic acidification led to SPI-2 gene activation remained unclear. In this study, we used super-resolution microscopy and determined that acid pH led to up-regulation of SsrA and SsrB, the main regulators of SPI-2 gene expression. Higher levels of *ssrB* transcripts have been previously observed using RT-PCR and microarrays comparing *Salmonellae* grown at different pH values (*Chakraborty et al., 2017*). SsrB levels were higher and showed larger intercellular variability

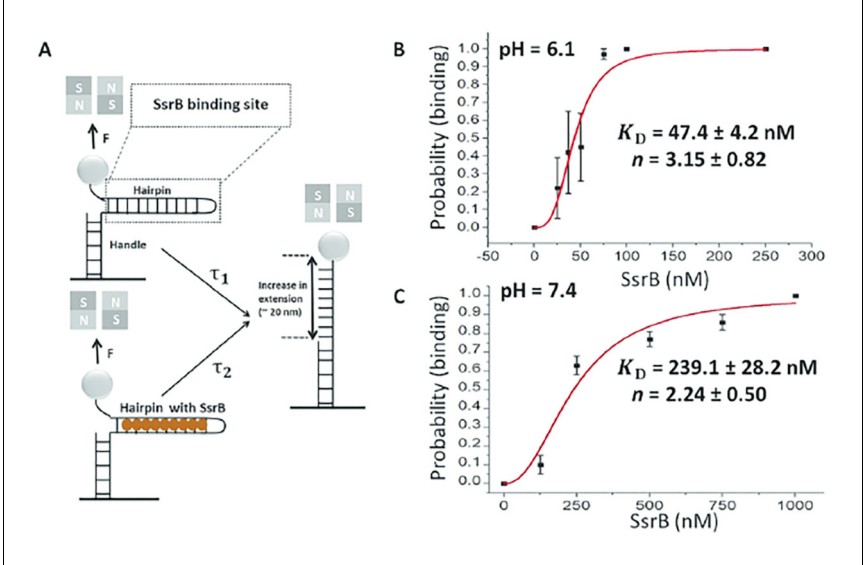

**Figure 8.** Quantification of SsrB-DNA binding affinity. (**A**) Schematic of the principle of the measurement (see *Gulvady et al., 2018* for more details). A force is applied to the hairpin using a pair of permanent magnets shown in gray (N: north pole, S: south pole). At a force slightly greater than the critical force, $F_c$, the naked DNA hairpin has a short lifetime of $\tau_1$, while the hairpin bound with SsrB has a much longer lifetime of $\tau_2$. This delayed unzipping indicates SsrB binding. See *Figure 8—figure supplement 1* for the binding traces. (**B**) The equilibrium binding probability as a function of SsrB concentration was plotted at pH 6.1 and (**C**) pH 7.4. These were the measured intracellular pH values of *Salmonella* in response to acid or neutral pH (*Chakraborty et al., 2015*; *Chakraborty et al., 2017*). The solid curves are the fitted curves to the Hill equation, the error bars represent the S.E.M. At pH 6.1 the $K_D$ was 47.4 ± 4.2 nM and the Hill coefficient (**n**) was 3.15 ± 0.82 nM. At pH 7.4, the $K_D$ increased to 239.1 ± 28.2 nM and the Hill coefficient (**n**) was still cooperative at 2.14 ± 0.5. Three to five independent tethers were analyzed for each point of the curves, and at least three independent probability values were determined at each SsrB concentration.

DOI: https://doi.org/10.7554/eLife.45311.032

The following source data and figure supplements are available for figure 8:

**Source data 1.**
DOI: https://doi.org/10.7554/eLife.45311.037
**Figure supplement 1.** Representative time traces of the extension change of the hairpin (pH = 6.1) in the absence of SsrB (**A**), and in the presence of SsrB at 37 nM (**B**) and 75 nM (**C**).
DOI: https://doi.org/10.7554/eLife.45311.033
**Figure supplement 1—source data 1.**
DOI: https://doi.org/10.7554/eLife.45311.034
**Figure supplement 1—source data 2.**
DOI: https://doi.org/10.7554/eLife.45311.036
**Figure supplement 2.** SsrB is a monomer at both acid and neutral pH.
DOI: https://doi.org/10.7554/eLife.45311.035

compared to SsrA during acid induction, corroborating previous reports that expression of *ssrA* and *ssrB* were uncoupled (*Feng et al., 2004*). Our observations that *ssrA* and *ssrB* were driven by separate promoters and their transcription was dependent on different global regulators, EnvZ/OmpR and PhoP/Q, further illustrates this point (*Figure 4—figure supplement 3*). This differential regulation is in contrast to many other two-component systems (e.g., OmpR/EnvZ) where the RR and the HK are translationally coupled (*Comeau et al., 1985*). Higher levels of SsrB, in the near absence of its kinase SsrA at neutral pH allows SsrB to drive a lifestyle switch (*Figure 1*, right), which up-regulates biofilm formation and establishes the carrier state (*Desai et al., 2016*). This asymptomatic state, in which *Salmonella* forms biofilms on gallstones in the gallbladder, allows *Salmonella* to persist in the environment and further transmit disease.

SsrB protein levels increased in response to acid pH, and SsrB binding to DNA in live *S*. Typhimurium cells also increased. SsrB binding was not clustered to specific regions of the chromosome (e.g the SPI-2 locus), but was distributed across the length of the cell (*Figure 3*), supporting the view that SsrB regulates many other genes that reside outside the SPI-2 locus (*Desai et al., 2016*). The SsrB-D56A non-phosphorylatable mutant maintained a low level of DNA binding above that of the K179A DNA binding mutant (*Carroll et al., 2009*), further supporting a role for unphosphorylated SsrB as a lifestyle switch in promoting biofilm formation (*Desai et al., 2016*). Acid-dependent DNA binding by SsrB was not a conserved feature of response regulators, because PhoP binding was not higher at acidic pH (*Figure 4—figure supplement 5C*) and OmpR binding only increased by 5% (*Figure 4—figure supplement 5F*), although the affinity of OmpR for DNA was acid-sensitive (*Chakraborty et al., 2017*). Even though PhoP protein levels increased during acid induction, higher levels of protein alone were not sufficient to drive DNA binding. Instead, our data suggests that PhoP is structurally less sensitive to changes in pH and is more reliant on phosphorylation by PhoQ to increase its affinity for DNA (*Lejona et al., 2004*). OmpR and PhoP are both members of the OmpR subfamily of response regulators with a winged-helix-turn-helix DNA binding domain (*Rhee et al., 2008*), indicating that structural homology does not necessarily determine functional homology. In contrast, SsrB is in the NarL/FixJ subfamily and its binding affinity was extremely pH-sensitive. Dimerization of SsrB was not pH-sensitive, according to size exclusion chromatography (*Figure 8—figure supplement 2*), thus understanding the effect of pH on SsrB will require more extensive analysis. NMR experiments are in progress to determine the precise molecular location of this pH-sensitivity switch. However, an examination of the pIs of NarL subfamily members indicates a range from 5.42 to 7.34 and suggests that SsrB (pI 7.34) is likely to be the most acid-sensitive compared to NarL (pI 5.6), for example. In response to in vitro acid stress, the *Salmonella* cytoplasm acidified to pH 6.1 (22). At this pH, SsrB would be positively charged, and this increase in positive charge would promote an interaction with negatively charged DNA. RcsB, with a pI of 6.85, is closest to SsrB; it may possess some acid-dependence. In our structural analysis, RcsB shared one DNA contact residue with SsrB, whereas NarL did not share any (*Carroll et al., 2009*). RcsB, like SsrB, can also act non-canonically in the absence of its cognate kinase (reviewed in *Desai and Kenney, 2017*).

## A caution regarding 'phosphomimetics'

Phosphoamino acids in proteins act as separate entities that diversify the chemical nature of protein surfaces. The phosphate group has a large hydrated shell and greater negative charge compared to Asp or Glu, whose carboxyl side chains have only a single negative charge and a smaller hydrated sphere (*Hunter, 2012*). However, a glutamic acid substitution is often used to replace aspartic acid with a claim that substitution produces a constitutively active protein. We examined the behavior of a 'constitutively active' SsrB variant (D56E), which was reported to be active in a mouse infection model (*Pérez-Morales et al., 2017*). Surprisingly, the D56E mutant was not able to activate *sseI* transcription (*Figure 6E*), nor was it capable of DNA binding in live cells (*Figure 6D*). One plausible explanation for the previous claim of SsrB 'activation' was that plasmid-expression of SsrB D56E (*Pérez-Morales et al., 2017*) led to inappropriate regulation (see *Haldimann and Wanner, 2001*). A vicinal pair of Glu residues might serve as a better phophomimetic than a single Glu, since this would generate a local double negative charge (*Pearlman et al., 2011*).

In summary, we have shown that acid pH is an important signal that regulates SsrB activity. Not only did the number of SsrA and SsrB molecules increase in acid pH compared to neutral pH, acid pH increased the affinity of SsrB for DNA and the percentage of SsrB bound to DNA increased substantially. Taken together, these factors work synergistically to increase SPI-2 gene expression in the SCV.

## Materials and methods

### Bacterial strains, media and culture conditions

Bacterial strains and plasmids used in this study are listed in Table 2. Overnight cultures of bacteria were grown in Luria-Bertani (Bacto, USA) broth at 37°C or 30°C, where appropriate, with shaking at 250 rpm. For antibiotic selection, the following antibiotics were used: Tetracycline (12.5 μg/ml), Ampicillin (100 μg/ml), Kanamycin (60 μg/ml) and Chloramphenicol (10 μg/ml). All antibiotics were

purchased from Sigma (USA). Growth in MgM media was based on previously published protocols (8,15). First, 1 ml of overnight culture was pelleted (6800 x g, 3 min), washed once with 1 ml of PBS and resuspended in 50 µl of PBS. Then, 3 ml of fresh MgM media pH 5.6 or pH 7.2 (8) was inoculated with 6 µl of the cell suspension. Cultures were then grown at 37°C with shaking at 250 rpm until $OD_{600}$ reached between 0.5-0.8.

## Molecular biology techniques

DNA manipulation techniques were performed as previously described (*Sambrook et al., 1989*) using appropriate restriction enzymes and DNA ligase (Thermofisher, USA). Gel extractions and plasmid extractions were performed using the QIAGEN Gel extraction and QIAGEN Miniprep kits, respectively, according to the manufacturer's instructions. Polymerase chain reaction (PCR) was performed using Q5 High-Fidelity DNA polymerase (NEB, USA) with primers (IDT, Singapore) listed in *Table 2* according to the manufacturer's instructions. PAGE purification was used for primers with lengths of more than 100 bps. Sanger sequencing was used to check the integrity of the DNA constructs used in this study (AIT Biotech, Singapore). Transformation of *Salmonella* strains was performed by standard electroporation protocols (*Sambrook et al., 1989*).

## Construction of *phoP* and *ompR* mutants

λ-Red recombination was used to construct the Δ*phoP* and the *phoP::kan* mutants in *Salmonella* (*Datsenko and Wanner, 2000*). Using plasmid pKD3 or pKD4 as the DNA template, a fragment containing the CmR or Kan resistance cassette flanked by 50 bps of DNA homologous to the regions immediately upstream and downstream of the *phoP* open reading frame was amplified by PCR using primers 110 and 111 and purified using the DNA gel extraction kit (Qiagen, USA). 1.5 µg of purified PCR product was then used to electroporate wild-type *Salmonella* containing plasmid pKD46. Electroporated cells were then recovered in 1 ml of LB broth at 30°C overnight with shaking before being plated onto LB plates containing 10 µg/ml of Chloramphenicol or 60 µg/ml of Kanamycin for overnight incubation at 37°C. To remove the CmR cassette from the *phoP::cmR* mutant, plasmid pCP20 was first introduced via electroporation into the mutant strain, cultured overnight in LB broth without antibiotics at 42°C then plated onto LB plates and grown overnight 37°C. Colonies that had lost the resistance cassette were identified by PCR and by the lack of growth in media containing chloramphenicol.

The *ompR::kan* strain was generated using primers pORed-H1-P1-f and EZed-H2-P2-r to generate the linear PCR fragment from pKD4 to be used for homologous recombination at the *ompB* locus. The plasmid pCP20 was used to remove the Kan resistance cassette to generate the Δ*ompR* strain.

## Construction of a C-terminal SsrB-PAmCherry fusion at its native chromosomal locus

λ-Red recombination (*Datsenko and Wanner, 2000*) was used to replace the native stop codon of *ssrB* open reading frames with a flexible linker-PAmCherry-TetRA fragment to generate the C-terminal SsrB-PAmCherry fusion at the native chromosomal locus in *Salmonella*. The SsrB-4-PAmCherry strain was constructed in two steps. First, a fragment containing the flexible 4XGGSG linker and PAmCherry open reading frame was amplified by PCR from genomic DNA isolated from the *E. coli* OmpR-4XGGSG-PAmCherry strain (*Foo et al., 2015*) using primers 1 and 3. Second, a fragment containing the *tetRA* resistance cassette was amplified by PCR from purified genomic DNA isolated from the *ssrA::tetRA Salmonella* strain (*Desai et al., 2016*) using primers 4 and 5. These two fragments were equally mixed and used as a template to amplify the 4XGGSG-PAmCherry-TetRA fragment which was flanked by 50 bps of DNA homologous to the regions immediately upstream and downstream of the *ssrB* stop codon. This PCR fragment was then purified and 2 µg of DNA was used for electroporation of wild-type *Salmonella* containing plasmid pKD46.

A similar strategy was used to construct the SsrB-10-PAmCherry strain with the exception that primer two was used for the first PCR amplification step to introduce the 10XGGSG linker. PCR and sequencing was used to confirm the correct integration of all the constructs at the appropriate chromosomal loci.

**Table 2.** Strains and plasmid used in this study.

| Strain number/plasmid | Description/genotype | Reference/Source |
|---|---|---|
| Salmonella strains | | |
| AL63 | Wild-type *Salmonella* Typhimurium 14028 s | Lab stock |
| AL142 | *ssrA::tetRA* (TetRA) | (*Winardhi et al., 2015*) |
| AL160 | *ssrA::tetRA attB:: ssrA-PAmCherry* (CmR) | This work |
| AL123 | *ssrB::ssrB-4-PAmCherry* (TetRA) | This work |
| AL125 | *ssrB::ssrB-10-PAmCherry* (TetRA) | This work |
| AL159 | *attB::* P*ssrB*-*PAmCherry* (CmR) | This work |
| AL60 | Δ*ssrB* | Lab stock |
| AL89 | Δ*ssrB attB::PAmCherry-4-ssrB* (CmR) | This work |
| AL92 | Δ*ssrB attB::PAmCherry-10-ssrB* (CmR) | This work |
| AL286 | Δ*ssrB attB::PAmCherry-4-ssrB D56A* (CmR) | This work |
| AL489 | Δ*ssrB attB::PAmCherry-4-ssrB D56E* (CmR) | This work |
| AL289 | Δ*ssrB attB::PAmCherry-4-ssrB K179A* (CmR) | This work |
| AL518 | *phop::kanR* | This work |
| AL522 | *phop::kanR attB::PAmCherry-4-ssrB* (CmR) | This work |
| AL325 | Δ*phoP* | This work |
| AL391 | Δ*phoP attB::phoP-4-PAmCherry* (CmR) | This work |
| AL394 | Δ*phoP attB::phoP-10-PAmCherry* (CmR) | This work |
| AL520 | *ompR::kanR* | This work |
| AL524 | *ompR::kanR attB::PAmCherry-4-ssrB* (CmR) | This work |
| AL525 | Δ*ompR* | This work |
| *E. coli* strains | | |
| AL53 | BW25141 pir+ | (*Bijlsma and Groisman, 2005*) |
| AL217 | BL21 (DE3) | Lab stock |
| Plasmids vectors | | |
| pKD46 | λ-Red recombinase expression plasmid (AmpR). | (*Kenney, 2018*) |
| pCP20 | Plasmid containing FLP recombinase for removal of CmR resistance marker flanked by FRT sites (AmpR). | (*Kenney, 2018*) |
| pKD3 | Plasmid used as a template for amplifying the CmR resistance cassette for constructing gene knockouts in Salmonella (CmR). | (*Kenney, 2018*) |
| pINT-ts | CRIM helper plasmid (AmpR). | (*Bijlsma and Groisman, 2005*) |
| pCAH63 | CRIM cloning vector (CmR). | (*Bijlsma and Groisman, 2005*) |
| pAL-4-PAmCherry | CRIM cloning vector containing the 4XGGSG linker upstream of PAmCherry (CmR). | This work |
| pAL-10-PAmCherry | CRIM cloning vector containing the 10XGGSG linker upstream of PAmCherry (CmR). | This work |
| pAL-PAmCherry-4-SsrB | CRIM plasmid containing the N-terminal PAmCherry-4XGGSG-*ssrB* fusion (CmR). | This work |
| pAL-PAmCherry-4-SsrB D56A | CRIM plasmid containing the N-terminal PAmCherry-4XGGSG-*ssrB* D56A fusion (CmR). | This work |
| pAL-PAmCherry-4-SsrB D56E | CRIM plasmid containing the N-terminal PAmCherry-4XGGSG-*ssrB* D56E fusion (CmR). | This work |
| pAL-PAmCherry-4-SsrB K179A | CRIM plasmid containing the N-terminal PAmCherry-4XGGSG-*ssrB* K179A fusion (CmR). | This work |
| pAL-P*ssrB*-PAmCherry | CRIM plasmid with a 500 bps 5'UTR fragment containing the *ssrB* promoter cloned upstream of PAmCherry. | This work |
| pAL-PAmCherry-10-SsrB | CRIM plasmid containing the N-terminal PAmCherry-10XGGSG-*ssrB* fusion (CmR). | This work |

*Table 2 continued*

| Strain number/plasmid | Description/genotype | Reference/Source |
|---|---|---|
| pAL-SsrA-10-PAmCherry | CRIM plasmid containing the C-terminal *ssrA*-10XGGSG-PAmCherry fusion (CmR). | This work |
| pAL-PhoP-4-PAmCherry | CRIM plasmid containing the C-terminal *phoP*-4XGGSG-PAmCherry fusion (CmR). | This work |
| pAL-PhoP-10-PAmCherry | CRIM plasmid containing the C-terminal *phoP*-10XGGSG-PAmCherry fusion (CmR). | This work |
| pMPM-A5Ω | pBR322 *ori araC* PBAD promoter Ω-interposon MCS (AmpR) | Lab stock |
| pMPM-A5Ω-his-ssrB | pMPM-A5Ω plasmid cloned with 6xhis-*ssrB* (AmpR) | (*Arnoldini et al., 2014*) |
| pMPM-A5Ω-his-ssrB-D56A | pMPM-A5Ω plasmid cloned with 6xhis-*ssrB* D56A (AmpR) | (*Arnoldini et al., 2014*) |
| pMPM-A5Ω-his-ssrB-D56E | pMPM-A5Ω plasmid cloned with 6xhis-*ssrB* D56E (AmpR) | This work |
| pMPM-A5Ω-his-ssrB-K179A | pMPM-A5Ω plasmid cloned with 6xhis-*ssrB* K179A (AmpR) | (*Chakraborty et al., 2017*) |
| pMPM-A5Ω-his-ssrBc | pMPM-A5Ω plasmid cloned with 6xhis-*ssrBc* (AmpR) | (*Arnoldini et al., 2014*) |
| pMPM-A5Ω-his-phoP | pMPM-A5Ω plasmid cloned with 6xhis-*phoP* (AmpR) | This work |
| pKF61 | Plasmid pKLC-II containing the promoter fragment of *sseI* fused to *lacZ* (AmpR) | (*Arnoldini et al., 2014*) |
| pKF8A | Plasmid pMC1871 containing the promoter fragment of *ssrB* fused to *lacZ* (TetRA) | (*Gao et al., 2017*) |

*Antibiotic resistance markers are expressed as follows: TetRA; tetracycline resistance, AmpR; ampicillin resistance, CmR; chloramphenicol resistance, KanR: Kanamycin resistance.

DOI: https://doi.org/10.7554/eLife.45311.039

## Construction of N-terminal SsrB fusions at the chromosomal λ*attB* site using a modified CRIM vector

Initial attempts to construct an N-terminal fusion at the native *ssrB* locus via the λ-Red counter-selection method (*Bochner et al., 1980*) were unsuccessful. To overcome this, we constructed two new CRIM vectors (pAL-4-PAmCherry and pAL-10-PAmCherry) for constructing C- or N-terminal PAm-Cherry fusions at the λ*attB* site in *Salmonella*. Both plasmids contain a multiple cloning site (EcoRI, KpnI, SalI, SphI and BamHI) upstream of a 4XGGSG or 10XGGSG flexible linker sequence that was linked with a PAmCherry fluorescent tag for creating C-terminal fusions. Additional restriction sites located immediately upstream (NdeI) and downstream (SacI, SpeI and SmaI) of the PAmCherry open reading frame can be used to construct N-terminal fluorescent fusions.

The pAL-4-PAmCherry plasmid was constructed in two steps. To introduce the multi-cloning site and 4XGGSG linker to PAmCherry, the PAmCherry gene was amplified by PCR (primers 24 and 26) using purified DNA isolated from the *E. coli* OmpR-PAmCherry (*Foo et al., 2015*) as template. Then, we amplified the pCAH63 backbone by PCR with primers 27 and 28, which simultaneously removes the plasmid 'stuffer' region (*Haldimann and Wanner, 2001*) and introduces an EcoRI restriction site to facilitate subsequent cloning steps. After digestion of both PCR fragments with EcoRI, the PAm-Cherry fragment was then ligated with the modified pCAH63 vector backbone to create pAL-4-PAm-Cherry. The same method was used for construction of the pAL-10-PAmCherry plasmid except that primer 25 instead of primer 24 was used to introduce the 10XGGSG linker.

Construction of the N-terminal PAmCherry-4-SsrB fusion was performed in two steps. First, we introduced the *ssrB* promoter region into the multi-cloning site immediately upstream of the PAm-Cherry open reading frame on plasmid pAL-4-PAmCherry. To do this, we amplified 500 bps of 5'UTR upstream of *ssrB* by PCR using primers 36 and 37 and cloned this fragment into plasmid pAL-4-PAmCherry using the EcoRI and NdeI restriction sites, resulting in the construction of plasmid pAL-P$_{ssrB}$-PAmCherry. Next, the *ssrB* open reading frame was amplified by PCR using primers 38 and 59 introducing the 4XGGSG linker to the 5' region upstream of the *ssrB* start codon. This fragment was cloned into plasmid pAL-P$_{ssrB}$-PAmCherry using SacI and SpeI restriction sites, creating pAL-PAmCherry-4-ssrB. The same steps were taken to construct the pAL-PAmCherry-10-ssrB construct, except that primer 39 instead of primer 38 was used to introduce the 10XGGSG linker. In addition to its use as a cloning vector, pAL-P$_{ssrB}$-PAmCherry was also introduced directly into wild-

type cells to create a *Salmonella* strain expressing PAmCherry alone for use in Spt-PALM experiments.

To construct plasmids expressing the N-terminal PAmCherry-SsrB D56A and K179A mutants, we amplified the entire *his-ssrB* orf containing the corresponding mutations from plasmids pMPM-A5Ω-his-ssrB-D56A (*Feng et al., 2004*) and pMPM-A5Ω-his-ssrB-K179A (*Carroll et al., 2009*) using primers 38 and 59 by PCR. After digesting the PCR fragments with SacI and SpeI, both fragments were cloned into plasmid pAL-P*ssrB*-PAmCherry digested with the same enzymes creating plasmids pAL-PAmCherry-4-SsrB D56A and pAL-PAmCherry-4-SsrB K179A, respectively. pAL-PAmCherry-4-SsrB D56E was constructed by first introducing the D56E mutation to wild-type *ssrB* with overlap-extension PCR using primers 38, 59, 202 and 203. The *ssrB* D56E-containing fragment was then cloned into plasmid pAL-P*ssrB*-PAmCherry using the SacI and SpeI restriction enzymes. All CRIM vector inserts were confirmed by sequencing and plasmids were maintained in *E. coli* BW25141, which encodes the *pir* gene required for CRIM plasmid replication (*Haldimann and Wanner, 2001*).

## Construction of the OmpR, PhoP and SsrA-PAmCherry C-terminal fusions at the chromosomal λ*attB* site using modified CRIM vectors

Construction of the various C-terminal PAmCherry fusions was as follows: For the OmpR-4-PAm-Cherry construct, a fragment containing 225 bp of 5'UTR and the entire *ompR* open reading frame was PCR amplified using primers pORst-E-f and PAmC_nt-B-r2and placed upstream of the 4XGGSG linker in plasmid pAL-4-PAmCherry using the EcoRI and BamHI restriction sites. PhoP-4-PAmCherry and PhoP-10-PAmCherry fusions were constructed in a similar manner. A fragment containing 270 bp of 5'UTR and the entire *phoP* orf was amplified by PCR using primers 114 and 115, digested with EcoRI and BamHI, and cloned upstream of the 4XGGSG or 10XGGSG linkers of plasmid pAL-4-PAm-Cherry or pAL-10-PamCherry, respectively. For the SsrA-10-PAmCherry fusion, we PCR amplified a fragment containing 500 bps of 5'UTR and the entire *ssrA* orf using primers 41 and 42, digested the fragment with EcoRI and SphI and cloned the fragment upstream of the 10XGGSG linker of plasmid pAL-10-PAmCherry digested with the same enzymes. All CRIM vector inserts were confirmed by sequencing and plasmids were maintained in *E. coli* BW25141 which encodes the *pir* gene required for CRIM plasmid replication (*Haldimann and Wanner, 2001*).

## CRIM vector integration

The protocol used for CRIM vector integration in *Salmonella* was modified from the *E. coli* integration protocol (*Zhou et al., 2004*). Overnight cultures of Salmonella strains containing the pINT-ts plasmid were first grown in LB media containing 100 ug/ml Ampicillin (Sigma, USA) at 30°C overnight with shaking at 250 rpm. The next day, 500 µl of overnight culture was added to 50 ml of fresh LB containing 100 µg/ml of Ampicillin in a 250 ml culture flask and grown for approximately 2.5 hr. The flask was then shifted to 42°C and grown for a further 30 min at the same shaking speed. Cells were then harvested by centrifugation (5000 x g 5 min) and washed three times with 20 ml of ice-cold sterile milliQ water. After the final wash, cells were resuspended in ~300 µl of ice-cold sterile milliQ water and 50 µl of the cell suspension was incubated with 2 µg of CRIM plasmid on ice for 5 mins. The cell-DNA suspension was then transferred into a pre-chilled 0.2 cm Micropulser electroporation cuvette (Biorad, USA) and electroporated with a voltage of 2.5 kV according to manufacturer's instructions. After electroporation, 1 ml of pre-warmed LB media was added to the cells and the cell suspension was then incubated at 42°C with shaking at 250 rpm for 40 mins to ensure efficient expression of the integrase gene and to simultaneously promote the loss of the pINT-ts plasmid. The cell suspension was further incubated overnight at 37°C and the next day, 200 µl of cells was plated onto LB agar containing 10 µg/ml of chloramphenicol (Sigma, USA) for overnight incubation at 37°C. Colonies that were present on plates were screened via PCR to ensure single-copy plasmid integration at the lambda *attB* site using primers 63, 64, 67 and 68 (*Haldimann and Wanner, 2001*). To avoid the co-expression of the native gene together with PAmCherry fusions from the integrated plasmid, corresponding CRIM plasmids were introduced into the appropriate *ssrB*, *ompR*, *phoP* and *ssrA::tetRA* deletion mutants in *Salmonella*.

## β-galactosidase activity measurement

β-galatosidase activity of the *sseI* and *ssrB* promoters was determined using previously published protocols (*Feng et al., 2003*; *Desai et al., 2016*). Bacterial strains containing plasmid pKF61 (*sseI-lacZ*) or pKF8A (*ssrB-lacZ*) were grown overnight in LB with appropriate antibiotics and then grown at 37°C in MgM pH 5.6 or pH 7.2 media until the $OD_{600}$ reached between 0.5–0.8. At this stage, 10–15 µl of culture was removed and placed into a well of a 96-well microtiter plate (ThermoFisher, China) containing 20 µl of chloroform (Sigma, USA) and 145 µl of lysis buffer (0.01% SDS, 50 mM β-mercaptoethanol in Z buffer) as described previously (*Feng et al., 2003*). To initiate color development, 30 µl of a 4 mg/ml ONPG (Sigma, USA) solution was added into each well. The β-galactosidase activity was represented in Miller Units and calculated as 1000 x [($OD_{420}$-1.75 x $OD_{550}$)]/t (min) x volume (ml) x $OD_{600}$). Measurements were made in a Tecan Infinite M200 plate reader and repeated at least twice in triplicates.

## Phage transduction

Phage transduction was performed as described previously (*Thierauf et al., 2009*) to introduce the PAmCherry constructs into different *Salmonella* deletion mutants. Briefly, 1 ml of an overnight culture of the donor strain was first lysed in 4 ml of phage broth (LB broth containing P22 phage, 0.2% glucose, 9.5 mM Citric acid, 0.78 mM $MgSO_4$, 75 mM $K_2HPO_4$, 26 mM $NaNH_4HPO_4$) at 37°C with shaking at 250 rpm. The supernatant was then collected using centrifugation (8000 x g 10 min) and mixed with 500 µl of chloroform (Sigma, USA). For transduction, 50 µl of phage lysate was mixed with 100 µl of overnight culture of the recipient strain for 1 hr at 37°C and plated onto LB plates containing 12.5 µg/ml Tetracycline. To confirm loss of the P22 phage, colonies were screened using green plates.

## Super-resolution microscopy (PALM)

Super-resolution imaging (PALM) was performed as previously described (*Foo et al., 2015*) with several modifications. Briefly, cells were first grown in MgM media to an $OD_{600}$ of 0.5–0.8 and fixed with 1.5% methanol-free paraformaldehyde (Wako Pure Chemicals, Japan) for 30 min. Cells were pelleted (6800 x g, 3 min), washed twice with PBS and then resuspended in 50 mM $NH_4Cl$ (Bright-Chem, Malaysia) in PBS for 2 hr to reduce excess paraformaldehyde. Cells were permeabilized with 1 mM EDTA (Sigma, USA) and 0.1% Triton X-100 (Sigma, USA) in PBS for 30 min followed by two PBS washes. Permeabilized cells were incubated with a 1:500 rabbit anti-*Salmonella* LPS (Abcam, USA) antibody-PBS solution for 1 hr to increase adherence of *Salmonella* to glass surfaces. Antibody-coated cells were then immobilized for 1.5 hr on 0.1% poly-L-lysine (Sigma, USA) coated-8-well glass chamber slides (Sarstedt, Germany) that were pre-cleaned with 3M KOH (Sigma, USA).

Imaging was performed on a Nikon N-STORM Super-Resolution microscope. *Highly inclined and laminated optical sheet* (HILO) illumination was done using a 561 nm laser line and a 405 nm laser line was used for PAmCherry activation. To image the PAmCherry fusions, between 8000–11000 frames were acquired at an exposure time of 100 msec per frame. Super-resolution imaging data was analyzed using rapidSTORM (*Wolter et al., 2012*). The localization precision determined by using the nearest neighbor based analysis (*Endesfelder et al., 2013*) was 15.0 ± 0.2 nm for PAmCherry. Localizations in consecutive frames that were present within a 40 nm radius were considered the same molecule and were treated as a single localization. A Gaussian blur of the mean localization precision was applied to the reconstructed images using ImageJ (*Schindelin et al., 2012*). 80 nm gold beads (BBI solutions, UK) were used as fiducial markers for drift correction at a 1:400 dilution in PBS. Where appropriate, PALM-PAINT imaging of membranes was performed as previously described (*Foo et al., 2015*), using 200 pM of Nile Red (Thermofisher, USA), after acquiring the PAmCherry PALM images.

To quantify levels of PAmCherry tagged molecules within cells, we first used the LocAlization Microscopy Analyzer (LAMA) program (*Malkusch and Heilemann, 2016*) to compute a localization-based image from the PALM coordinate list generated with rapidSTORM using a desired pixel size of 10 nm and a 255 maximum intensity value. Thus, one gray scale is added to the respective pixel for each localization. Brightfield images were manually analyzed with ImageJ and used to define cell boundaries (cell area in $\mu m^2$). To obtain the number of localizations (# of molecules) for individual cells, the integrated intensity (RawIntDen) within the determined cell boundaries of each cell was

measured in the localization-based LAMA image using ImageJ. Because pH affects cell length, and the number of molecules per cell is affected by cell length, a more appropriate representation of our data involved normalizing the number of molecules by area. Hence, the number of molecules/$\mu m^2$ value was calculated by normalizing the integrated intensity to the respective cross-sectional cell area. Box plots, averages and standard deviations were obtained using OriginPro software (Origin-Lab, USA).

## Spt-PALM

Cells were grown in MgM media to an $OD_{600}$ of 0.5–0.8, concentrated by centrifugation (6800 x g, 3 min), then placed onto a 2% agarose pad containing MgM pH 7.2 or pH 5.6 media and subsequently sealed with a clean glass coverslip. Agarose pads were prepared by pipetting 60 µl of the molten agarose solution into the center of a 65 µl gene frame (Thermofisher, USA) adhered onto a 76 × 26×1 mm glass slide (Marienfeld, Germany) and immediately covering the agarose surface with a clean 22 × 22 mm glass coverslip (High Precision, Germany). Coverslips were cleaned overnight in 3M KOH (Sigma, USA) solution, followed by two 30 min cycles of sonication in a S60H ElmaSonic waterbath sonicator (Elma, USA). After drying the coverslips overnight, the coverslips were finally plasma-cleaned for 30 min in a plasma cleaner (Harrick plasma, USA) to reduce background fluorescence prior to use.

Spt-PALM experiments were performed using similar settings as SMLM, except that 50,000 frames were acquired at an exposure time of 15 ms, resulting in ~17 ms per frame. Single molecule signals (spots) were detected and connected using the Fiji tracking plugin TrackMate. Spots were linked to form a track using a maximum linking distance of 0.7 µm. Only tracks with more than five spots were used for data analysis. Tracks were then further analysed as described (*Gao et al., 2017*).

The displacement $r$ can be calculated as the distance the molecule travelled in one camera frame. The distribution of $r$ across all tracks can be plotted as a histogram with the probability distribution function (PDF) given as (*Yang et al., 2016*):

$$\text{PDF}(r,\tau) = \frac{r}{2D\tau}\exp\left(-\frac{r^2}{4D\tau}\right) \tag{1}$$

where $D$ is the diffusion coefficient of the molecule and $\tau$ the time between each frame. In theory, fitting the histogram of displacement $r$ with *Equation 1* will give the value of $D$. However, the fitting depends on the bin size of the histogram, which can affect the fitting result. Hence, the cumulative distribution function (CDF) of displacement $r$ is used instead:

$$\text{CDF}(r,\tau) = \int_o^r \text{PDF}(r,\tau)\mathrm{d}r = 1 - \exp\left(-\frac{r^2}{4D\tau}\right) \tag{2}$$

For molecules undergoing multiple diffusion states, a linear combination of CDF with multiple $D$ can be used (*Yang et al., 2016*; *Chen et al., 2015*; *Gebhardt et al., 2013*). A three-component diffusion model best fitted our data (*Figure 6—figure supplement 1*):

$$\text{CDF}(r,\tau) = 1 - \left[ F_1\exp\left(-\frac{r^2}{4D_1\tau}\right) + F_2\exp\left(-\frac{r^2}{4D_2\tau}\right) + F_3\exp\left(-\frac{r^2}{4D_3\tau}\right) \right] \tag{3}$$

where $F_n$ is the relative frequency of the different diffusion states $D_n$.

We performed Spt-PALM on fixed cells expressing SsrB-PAmCherry (cells grown in pH 5.6, fixed with 1.5% PFA). Due to the localization error of each spot, the immobile molecule is apparently moving. The CDF was fitted with *Equation 3* to obtain $D_1 = 0.020 \pm 0.002$ $\mu m^2 s^{-1}$ ($F_1 = 26.9 \pm 4.0\%$), $D_2 = 0.066 \pm 0.003$ $\mu m^2 s^{-1}$ ($F_2 = 65.0 \pm 4.0\%$) and $D_3 = 0.88 \pm 0.08$ $\mu m^2 s^{-1}$ ($F_3 = 8.1 \pm 0.6\%$). The values of $D_1$ and $D_2$ were very small and represented 91.9% of the population. The presence of two low $D_1$ and $D_2$ values was most likely due to the different population of immobile spots with a different signal to noise ratio, leading to different localization precisions. Sometimes during the linking of the spots to form a track, a spot belonging to one molecule can be linked to a different molecule. This happens when the final spot from the first molecule photobleaches, and a spot from another molecule appears at a different location, but still within our maximum linking distance. The 8.1% that has a higher $D_3$ is due to such linking errors. We then set a displacement threshold $r_0 = 0.127$ µm, below which 91.9% of the population (from $F_1$ and $F_2$) were included. Thus, 91.9% of the population

has a value of $D_1$ and $D_2$ with an $r$ value less than 0.127 µm. This was used to assign $D_2$ to a transient weaker binding form of SsrB (see *Figure 6*).

## SseB immunofluorescence

Visualization of the SseB translocon was based on a previously published protocol (*Chakraborty et al., 2015*). *Salmonella* wild-type, ΔssrB and the PAmCherry-4-SsrB expressing strains were first grown in acid pH MgM media and prepared as described for PALM imaging. Then, the cells were incubated with a 1:500 dilution of rabbit anti-SseB primary antibody in PBS buffer containing 2% BSA and 0.1% Tween for 1 hr. After washing the cells five times by centrifugation with PBS, the cells were incubated with a 1:500 dilution of donkey anti-rabbit Alexa 488 conjugated secondary antibody (Thermofisher scientific, USA) in the same PBS buffer. Cells were then washed five times with PBS before being placed on 2% agarose pads and sealed with KOH-cleaned coverslips for microscopy. A 470 nm laser line was used to first image SseB fluorescence before acquiring the SsrB-PAmCherry signal using PALM.

## Overexpression and purification of proteins

*E. coli* BL21 (DE3) was used as a host for overproduction of SsrB, OmpR and PhoP proteins used for Atomic force microscopy and in vitro transcription assays. Detailed procedures for their purification have been previously described (*Desai et al., 2016*; *Feng et al., 2004*; *Walthers et al., 2007*; *Chakraborty et al., 2017*; *Carroll et al., 2009*; *Castelli et al., 2000*). (*Feng et al., 2004*. The PhoP-His expressing plasmid was constructed by PCR amplification of *Salmonella* genomic DNA using primer pair DW772 and DW773, TOPO cloning into pCR2.1 as described by the manufacturer (Invitrogen) and subsequent cloning into pMPMA5Ω using *Eco*RI and *Xba*I restriction sites.

## Atomic force microscopy

The DNA used for AFM was generated by amplifying a 704 bp fragment containing the *sifA* promoter by PCR (primers 195 and 196) from purified genomic DNA prepared from wild-type 14028 s *S*. Typhimurium and gel purified. The AFM experiments were performed on glutaraldehyde-coated mica surfaces according to the previously described method (*Desai et al., 2016*) with slight modifications. 20 ng DNA was mixed with an appropriate concentration of SsrB in a 100 µl reaction and incubated for 15 min at room temperature in either neutral (50 mM KCl, 50 mM Tris-HCl pH 6.8) or acidic buffer (50 mM KCl, mM MES pH 6.1). The DNA:protein mixture was deposited on the glutaraldehyde- modified mica for 15 min. The sample was gently washed with deionized water, and then dried with $N_2$ gas. 2 × 2 micron images were acquired with a resolution of 1024 × 1024 on a Bruker Dimension FastScan AFM system using the tapping mode with a silicon nitride cantilever (FastScan A, Bruker). Raw AFM images were processed using Gwyddion software (http://gwyddion.net/). After background subtraction (contributed by the mica surface), the color range of images was set from 0 to 1.5 nm, the images were then saved. The relative height values were exported as an ASCII file. The relative height distribution histogram of *sifA* promoter complexes was plotted using >8 images, each image contains approximately 100 DNA molecules. The values were exported as an excel file and plotted by GraphPad Prism. The experiments were prepared in duplicate (i.e., 16 images total/ sample).

## Single molecule unzipping assay

The DNA hairpin assay was prepared as described (24). The hairpin consisted of a 20 basepair stem of a 2X repeat of the high-affinity SsrB binding site in the *csgD* promoter (15). Using magnetic tweezers, a controlled force was applied on the hairpin such that it instantly transitioned from a closed, double-stranded (ds) DNA state to an open, single-stranded (ss) DNA state (See Figure 8-figure supplement 1). A delay in the opening of the hairpin was observed when SsrB was bound. By quantifying this delay, we assess the strength of SsrB binding to the hairpin for a range of SsrB concentrations and calculated the dissociation constant ($K_D$) of binding. Three to five tethers were analyzed for each point of the binding curves.

## Nucleoid size measurement

Cells were grown in MgM pH 5.6 or pH 7.2 and fixed with 1.5% PFA for 30 mins. They were then pelleted and permeabilized with 0.1% Triton X100 in PBS for 30 mins. The cells were washed three times with PBS via centrifugation, followed by staining with DRAQ5 for 15 mins. They were then washed 3 times with PBS and placed onto a 2% agarose pad and subsequently sealed with a clean glass coverslip. Imaging was performed using structured-illumination microscopy (SIM) on a W1 spinning Disk microscope (CSU-W1 Nikon, Japan) combined with the Live-SR system (Roper scientific) and equipped with a Plan-Apo $\lambda$ 100x oil objective (1.45 NA, Nikon, Japan) as previously described (*Gao et al., 2017*). The area of the nucleoid per cell was quantified as previously described (*Gao et al., 2017*).

## Acknowledgements

We are grateful to Mike Heilemann and Christoph Spahn (Goethe University) for helpful discussions, Tony Hunter (Salk Institute) for discussions of phosphomimetics and Stuti Desai (Mechanobiology Institute) for the *ssrA:tetRA* construct. Melanie Lee in the MBISciCom team created *Figure 2* and Diego Pitta de Araujo edited *Figure 1*.

## Additional information

### Funding

| Funder | Grant reference number | Author |
| --- | --- | --- |
| National Institutes of Health | AI-123640 | Linda J Kenney |
| Veteran's Affairs | IOBX-000372 | Linda J Kenney |
| Ministry of Education - Singapore | | Linda J Kenney |
| Ministry of Education | MOE2018-T2-1-038 | Linda J Kenney |

The funders had no role in study design, data collection and interpretation, or the decision to submit the work for publication.

### Author contributions

Andrew Tze Fui Liew, Conceptualization, Investigation, Writing—original draft, Writing—review and editing; Yong Hwee Foo, Data curation, Investigation, Methodology, Writing—original draft, Writing—review and editing; Yunfeng Gao, Formal analysis, Investigation, Writing—original draft, Writing—review and editing; Parisa Zangoui, Investigation, Writing—original draft, Writing—review and editing; Moirangthem Kiran Singh, Investigation, Methodology, Writing—original draft, Writing—review and editing; Ranjit Gulvady, Formal analysis, Investigation, Methodology, Writing—original draft, Writing—review and editing; Linda J Kenney, Conceptualization, Funding acquisition, Methodology, Writing—original draft, Project administration

### Author ORCIDs

Andrew Tze Fui Liew https://orcid.org/0000-0003-4792-9637
Yong Hwee Foo https://orcid.org/0000-0003-2337-469X
Moirangthem Kiran Singh https://orcid.org/0000-0001-9620-5454
Linda J Kenney https://orcid.org/0000-0002-8658-0717

### Decision letter and Author response

Decision letter https://doi.org/10.7554/eLife.45311.042
Author response https://doi.org/10.7554/eLife.45311.043

## Additional files

### Supplementary files
• Transparent reporting form
DOI: https://doi.org/10.7554/eLife.45311.040

### Data availability
All data generated or analysed during this study are included in the manuscript and supporting files. Source data files have been provided for all figures.

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
