## [Decision Letter]

Thank you for submitting your article "Single cell, super-resolution imaging reveals acid-dependent SPI-2 gene regulation by SsrB" for consideration by *eLife*. Your article has been reviewed by three peer reviewers, and the evaluation has been overseen by a Reviewing Editor and Gisela Storz as the Senior Editor. The following individuals involved in review of your submission have agreed to reveal their identity: James Slauch and Alexander Westermann..

The reviewers have discussed the reviews with one another and the Reviewing Editor has drafted this decision to help you prepare a revised submission.

Summary

In the present study, you used super-resolution microscopy approaches to show that SsrA and SsrB, comprising the two component regulatory system that controls expression of the SPI2 T3SS in *Salmonella*, are acid inducible and that there is pH-dependent stimulation of DNA binding by SsrB. The work builds on your previous finding that *Salmonella* cytoplasm acidifies in the macrophage phagocytic vacuole, which is critical for induction of virulence factors. The reviewers all agreed that the findings are novel and relevant for our understanding of *Salmonella* pathogenesis, and potentially for broader response regulators. They deem the results convincingly supported by the experimental data, but request to see a few more experimental additions to your work and stylistic changes before it can be accepted for publication.

Essential revisions:

1) Characterise the complemented phoP and ompR mutants in Figure 3.

2) Measure the intracellular bacterial pH in the experimental conditions used here to inform how the extracellular pH affects the intracellular pH, for example where SsrB binding is tested.

3) Characterise how acid pH affects the conformation or possibly the dimerization of SsrB. How does a ssrB mutant that cannot dimerise behave regarding binding to the SPI-2 promoter at low pH?

4) In addition to PhoP and OmpR here tested, test other NarL transcriptional factors binding ability at low pH.

5) Add statistical analysis wherever possible especially in Figures 2B/ 3B/ 4B/ 3D/ 6D.

6) Back Figure 6 results with EMSA experiments.

7) Document what is the background signal obtained in Figure 2 when there is no PAmCherry expressed by the bacteria (in order to state that SsrB can be detected in a phoP mutant).

8) Annotate figures more extensively, particularly Figure 6.

---

## [Author Response]

Essential revisions:1) Characterise the complemented phoP and ompR mutants in Figure 3.

These results are now included in the Figure 4—figure supplement 3.

2) Measure the intracellular bacterial pH in the experimental conditions used here to inform how the extracellular pH affects the intracellular pH, for example where SsrB binding is tested.

We have already made this measurement and published the result (Chakraborty, et al., 2017). In the present manuscript, we clearly stated: “At acid pH (pH 6.1, the intracellular pH we measured previously (Chakraborty et al., 2017))”, …. We added a sentence in the legend: “The pH values were selected based on the measured values in response to acid stress (Chakraborty et al., 2017)”.

3) Characterise how acid pH affects the conformation or possibly the dimerization of SsrB.

We analyzed SsrB by gel filtration and dimerization is not pH-dependent (see Author response image 1). A sentence to this effect has been added to the manuscript (subsection “Acid pH increases SsrB binding to DNA”).

7.4 Buffer: 20 mM Hepes pH 7.4, 50 mM KCl, 2 mM MgCl2, 2 mM DTT, 5% glycerol; 6.1 Buffer: 20 mM MES pH 6.1, 50 mM KCl, 2 mM MgCl2, 2 mM DTT, 5% glycerol. Thus, SsrB is a monomer at both acid and neutral pH.

How does a ssrB mutant that cannot dimerise behave regarding binding to the SPI-2 promoter at low pH?

In our previous study, we showed that a dimerization mutant no longer binds to DNA (Carroll et al., 2009). If the protein already does not bind to DNA, we cannot measure an effect of acid pH on DNA binding.

4) In addition to PhoP and OmpR here tested, test other NarL transcriptional factors binding ability at low pH.

We respectfully disagree with the reviewers on this point and feel that this is way beyond the scope of our manuscript. We have tested PhoP and OmpR because we also study these regulators, have over-expression vectors and purified proteins readily available (and because they regulate SsrB!). To demand other NarL family members really is not the point of the manuscript and frankly was the type of point I thought was avoidable by the review strategy of *eLife*. Of course, we are interested in the molecular basis of the acid-dependent conformational change in SsrB, and we are just beginning to attempt some NMR studies. It will likely be difficult, because of solubility problems with the full-length protein, etc. Assuming we can overcome these problems, we will then determine the chemical shift changes that occur in acid pH. Once we establish the molecular basis of the pH dependence with SsrB, we will see whether any of these amino acid residues are conserved in NarL family members. I rather doubt it, though, because we did an extensive analysis of homologues when we solved the structure of SsrBc (Carroll et al., 2009) and they are NOT conserved. I also rather doubt there is conservation, because NarL, for example, requires phosphorylation to bind to DNA (RE Dickerson, Biochem. 1996) because the recognition helix is physically blocked by the N-terminus and phosphorylation re-orients the interface, which is not the case with SsrB. We have, however, examined the pIs of several family members and include a discussion about this feature in an additional paragraph in the Discussion (subsection “Acid pH increased SsrA/B levels and SsrB affinity for DNA”).

5) Add statistical analysis wherever possible especially in Figures 2B/ 3B/ 4B/ 3D/ 6D

We assume you mean 4C. We have added the analysis to the figure legends, and include * indicating the p value in the figures.

6) Back Figure 6 results with EMSA experiments.

We have already performed extensive EMSA analysis with SsrB (Carroll et al., 2009). EMSAs and most other DNA binding assays, including fluorescence anisotropy, are exquisitely pH-sensitive. That is why we used AFM. At acid pH, the complexes will not migrate properly through the gel, and titrating them just before loading will disturb the equilibrium. We also believe that actually visualizing DNA binding by AFM is more informative than looking at a band shift. It is evident from recently published papers (e.g., Perez-Morales et al., 2017) that even at 2 μm SsrBc there was still free DNA apparent, yet in our submitted manuscript, we measured a Kd of 47 nM (!), raising doubts as to how meaningful their published EMSAs are. We were also able to perform these experiments on full length protein, rather than just the C-terminal domain (SsrBc). This is significant, because the pI of SsrBc is 9.36 (!) compared to a pI of 7.34 for the full-length protein.

7) Document what is the background signal obtained in Figure 2 when there is no PAmCherry expressed by the bacteria (in order to state that SsrB can be detected in a phoP mutant).

We had already performed this experiment, but we now include the results. We imaged the D ssrB strain (without PAmCherry) to get the levels of background signals. Average localization counts for the strain lacking PAmCherry was 4 localizations/um2 (n = 66 cells,) while the PAmCherry-SsrB *phoP::kan* strain was 16 localizations/um2, so it is clear that we can detect SsrB above background. This information is included in the legend to Figure 4. A description of these results is included in subsection “Acid pH increases SsrB and SsrA levels” (and the new figure is Figure 4—figure supplement 2). The old Supplementary figures 2-9 are now increased.

8) Annotate figures more extensively, particularly Figure 6.

Done. We have now added some clarifying sentences, both in the legend and in the text.